# Frequency spectra and the color of cellular noise

Ankit Gupta [1] & Mustafa Khammash [1✉]

The invention of the Fourier integral in the 19th century laid the foundation for modern spectral analysis methods. This integral decomposes a temporal signal into its frequency components, providing deep insights into its generating process. While this idea has precipitated several scientific and technological advances, its impact has been fairly limited in cell biology, largely due to the difficulties in connecting the underlying noisy intracellular networks to the frequency content of observed single-cell trajectories. Here we develop a spectral theory and computational methodologies tailored specifically to the computation and analysis of frequency spectra of noisy intracellular networks. Specifically, we develop a method to compute the frequency spectrum for general nonlinear networks, and for linear networks we present a decomposition that expresses the frequency spectrum in terms of its sources. Several examples are presented to illustrate how our results provide frequency-based methods for the design and analysis of noisy intracellular networks.

[1] Department of Biosystems Science and Engineering, ETH Zürich, Mattenstrasse 26, 4058 Basel, Switzerland. ✉email: mustafa.khammash@bsse.ethz.ch

Modern microscopy and the advent of a wide array of fluorescent proteins[1] have afforded scientists the unprecedented ability to monitor the dynamics of living biological cells[2]. The rapid pace of development in imaging technology coupled with advanced image processing techniques has made it viable to obtain high-resolution time-lapse live-cell data for a multitude of cell-types and biological processes. Recent innovations in microfluidics make it possible to quantitatively measure single-cell dynamics for long periods of time over multiple generations[3–5]. These trends underscore the need for developing theoretical and computational tools that are specifically geared towards quantitatively extracting information about intracellular networks from live single-cell imaging data. One of the main reasons why the development of such tools is mathematically challenging is that the dynamics of single-cells is inherently noisy due to randomness in molecular interactions that constitute intracellular processes, and hence single-cell dynamics must be described with stochastic models that are more difficult to analyse than their deterministic counterparts[6]. These stochastic models usually represent the reaction dynamics as a continuous-time Markov chain (CTMC) and the existing methods for analysing them have mostly focussed on solving the chemical master equation (CME) that governs the evolution of the probability distribution of the random state[7]. While these methods have been successfully applied in several significant biological studies[8,9], they typically do not account for temporal correlations in time-traces of living cells, but rather they are designed to connect network models to flow-cytometry data[10] where temporal correlations are anyway lost due to discarding of the measured cells. Temporal correlations are a feature of single-cell trajectories that contain valuable information about the underlying network, and in order to access this information we need computational methods that can efficiently deduce the temporal correlation profile from a given stochastic reaction network model.

As is well-known in engineering and physics communities among many others, frequency-domain analysis is a powerful way to analyse random signals and systematically study temporal correlations. In particular, a signal's power spectral density (PSD) measures the power content at each frequency, and it is related to the signal's temporal autocovariance function via the Fourier Transform (see Box 1). The PSD of a single-cell trajectory is intimately related to the underlying network's architecture and parametrisation within the observed cell[11]. There exist many studies that have successfully unravelled this relationship and discovered mechanistic principles for specific examples of reaction networks. For example, in ref. [12] the role of feedback-induced delay in generating stochastic oscillations is explored and in ref. [13]

---

**Box 1 | Frequency domain analysis of stochastic signals**

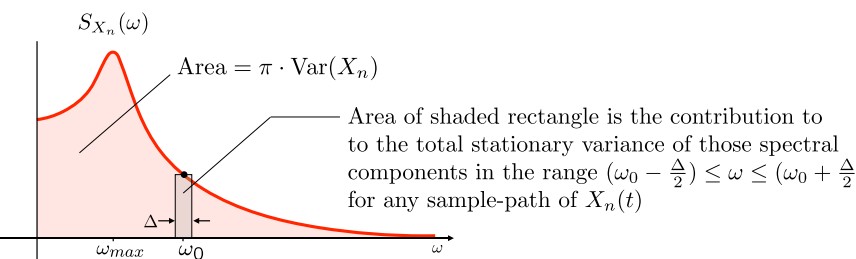

Consider a reaction network, comprising species $\mathbf{X}_1, ..., \mathbf{X}_d$ whose copy-number dynamics is described by an ergodic continuous-time Markov chain (CTMC) $(X(t))_{t \geq 0}$ with stationary distribution $\pi$. Our goal is to estimate the PSD which measures the strengths of oscillatory components of various frequencies in the output signal $(X_n(t))_{t \geq 0}$ tracking the copy-number trajectory for species $\mathbf{X}_n$. We first subtract the stationary mean $\mathbb{E}_\pi(X_n)$ and construct the mean zero signal as $\tilde{X}_n(t) = X_n(t) - \mathbb{E}_\pi(X_n)$ and then the time-averaged signal power $P(X_n)$ is equal to the stationary variance $\mathrm{Var}_\pi(X_n)$, i.e.

$$P(X_n) := \lim_{T \to \infty} T^{-1} \int_0^T \left( \tilde{X}_n(t) \right)^2 dt = \mathrm{Var}_\pi(X_n).$$

The power spectral density (PSD) for the output signal is given by

$$S_{X_n}(\omega) = \lim_{T \to \infty} T^{-1} |\mathcal{F}_T(\omega)|^2, \text{ where } \mathcal{F}_T(\omega) = \int_0^T \tilde{X}_n(t) e^{-i\omega t} dt$$

is the one-sided Fourier Transform, $\omega$ is the frequency and $i = \sqrt{-1}$. This PSD is related to the autocovariance function

$$R_{X_n}(\tau) := \mathbb{E}\left[ \tilde{X}_n(t) \tilde{X}_n(t + \tau) \right] = \lim_{T \to \infty} T^{-1} \int_0^T \tilde{X}_n(t) \tilde{X}_n(t + \tau) dt$$

by the well-known Wiener-Khintchine Theorem[85] that shows that the PSD can be expressed as the two-sided Fourier Transform of the autocovariance function

$$S_{X_n}(\omega) = \int_{-\infty}^{\infty} R_{X_n}(\tau) e^{-i\omega\tau} d\tau. \tag{1}$$

The interpretation of the PSD curve is given above. The location $\omega_{max}$ of its global maximum is considered to be the oscillatory frequency of the output signal.

Commonly the PSD is estimated by first sampling a discrete time-series from a simulated CTMC trajectory at steady-state, and then taking its discrete Fourier transform (DFT) to estimate $\mathcal{F}_T(\omega)$ which then yields the PSD. This nonparametric procedure for PSD estimation is often called the periodogram method and it has known drawbacks due to estimator bias and inconsistency that often manifests in a high variance of the PSD estimator. The reliability of the estimator can be improved by ensemble averaging, windowing or artificial smoothing[32], but the underlying problems that compromise the accuracy of the PSD estimate still remain.

a stochastic amplification mechanism for oscillations is found. Notably, the exact PSD for linear reaction networks was derived in ref. [14] and this was used to show how in gene expression networks post-translational modification reaction reduces the noise by serving as a low-pass filter.

Other works in this direction have relied on approximating the CTMC with a stochastic differential equation (SDE) such as the linear noise approximation (LNA)[15] or the chemical Langevin equation (CLE)[16]. With these SDE-based approaches the protein PSD for gene-regulatory networks was investigated in refs. [17–19], the relationship between input and output PSD for a single-input single-output system was computed in ref. [20], the single-cell PSD for a general biomolecular network in the vicinity of a deterministic Hopf bifurcation was determined in ref. [21] and corrections to the LNA-based PSD estimates were systematically derived in ref. [22]. Even though SDE approximations make the problem of computing the PSD analytically tractable, their accuracy is severely compromised if any of the species are in low copy-numbers, as is the case for many synthetic networks where low copy-numbers are desired in order to reduce metabolic load on the host cell[23]. Moreover, even when the species copy-numbers are uniformly large, the accuracy of SDE approximations can only be guaranteed over finite time-intervals[24], and hence the PSD, which is estimated at steady-state, could have an error (see the example presented in Fig. 4). It must be noted that for linear networks these approximations yield the exact PSD but if the network has nonlinear propensities then the error in the derived PSD expression can be significant[25]. In order to address these issues, we need PSD estimation methods that work reliably for CTMC models, especially in the low copy-number regime, without requiring any dynamical approximations. The aim of this paper is to develop such a method.

In a recent paper[26], the analytical relationship between the PSDs of the output species and its time-dependent production rate was derived for CTMC models of certain reaction networks including birth-death and simple gene expression. While this analysis enables investigation of the dynamics of the protein creation process from experimentally measured protein time-traces, it does not extend to nonlinear networks, such as gene expression networks with transcriptional feedback, for which some analytical results exist for simplified models[27].

A recurring theme in the existing literature is that typically the autocovariance function is well-approximated by the sum of a few exponential functions[18,20,26], and consequently the PSD is a rational function of a special form. This low dimensional feature can be theoretically explained by appealing to the compactness of the resolvent operator[28] associated with the CTMC, which as we prove, is connected to the PSD. Exploiting this connection we develop the multipoint Padé approximation[29] technique for estimating the PSD for a general nonlinear stochastic reaction network. This method, which we refer to as *Padé PSD*, computes the PSD expression based on certain stationary expectations. We design efficient Monte Carlo estimators to estimate the required expectations by generating a handful of simulations of an augmented CTMC, constructed by adding certain state-components and reactions to the original CTMC. We show how this augmented CTMC construction not only facilitates PSD estimation but also its empirical validation.

Our PSD estimation approach is semi-analytic, in the sense that analytical expressions for the PSD are found by first estimating certain quantities with simulation. Such approaches have become increasingly popular in recent years, as they provide viable solutions to nonlinear problems which are otherwise analytically intractable[30]. Analytical expressions for the PSD are known in the special case of linear reaction networks[14], where all reaction propensity functions are affine functions of the state

variables. We show how this expression can be alternatively derived via the resolvent connection and we also generalise this result to allow for arbitrary time-varying inputs. This generalisation yields a PSD decomposition result that is similar to what was found in previous SDE-based studies[20] and it extends the recent results in ref. [26].

Given a stochastic reaction network model, commonly the single-cell PSD is estimated with nonparametric methods by first simulating a trajectory, and then sampling it at finitely-many timepoints to obtain a discrete time-series whose PSD can be straightforwardly computed with the Discrete Fourier Transform (DFT)[31]. Either one can apply the DFT directly to the time-series to estimate the PSD or one can first estimate the autocovariance function and then compute its DFT (see Box 1 for more details). While the latter approach is computationally very expensive due to the autocovariance function computation, the former approach yields an inconsistent estimator for the PSD, which implies that the estimator variance does not vanish, even as the time-series length tends to infinity. To mitigate this inconsistency issue, PSDs from several independent trajectories are averaged, at the cost of significant computational burden as trajectory simulations are time-consuming. More importantly, the averaged PSD may still not be accurate because it is based on discrete sampling of continuous signals that can cause the problem of aliasing which distorts the estimated PSD by introducing frequency components corresponding to the sampling operation (see Chapter 1 in ref. [32]). As shown by the Nyquist's Sampling Theorem[33] we can mitigate this aliasing effect by choosing the time-step parameter that is smaller than half of the reciprocal of the maximum frequency represented in the signal. However, for stochastic dynamics this criterion is unusable as the range of frequencies in the signal is very wide and picking a very small time-step can lead to computational intractability. These issues motivated us to devise Padé PSD that is not based on discrete-sampling and provides a parametric approach for estimating the PSD that rather than relying on only the output signal, uses full information contained in the stochastic model of the dynamics.

We illustrate our results with applications of relevance to both systems and synthetic biology. Using our PSD decomposition result for linear networks, we demonstrate how PSDs enable differentiation between two fundamental types of adapting circuit topologies, viz. Incoherent Feedforward (IFF) and Negative Feedback (NFB)[34], in the presence of dynamical intrinsic noise. We also present an example where the phenomenon of single-cell entrainment is examined in the stochastic setting using our PSD decomposition result. Employing Padé PSD we illustrate how the performance of certain synthetic circuits, with noisy dynamics, can be optimised. Specifically, we examine the problem of optimising the oscillation strength of a well-known synthetic oscillator (called the repressilator[35]) and the problem of reducing single-cell oscillations which can arise when an intracellular network is controlled with the antithetic integral feedback (AIF) controller[36] that has the important property of ensuring robust perfect adaptation despite randomness in the dynamics and other environmental uncertainties. Lastly, we present examples to highlight how our Padé PSD method helps in the study of oscillations caused by cell-division cycles as well as facilitate parameter inference from experimentally measured single-cell trajectories, by providing clean and accurate estimations of the PSD. Interestingly, inferring a parameter with PSD does not require the explicit knowledge of the proportionality constant that relates the measured signal to the copy-number of the output species[37].

## Results

**The stochastic model.** We first describe the CTMC model for a reaction network and define the resolvent operator associated with

with it. We then connect this operator to the PSD. This connection shall be exploited later to develop our analytical and computational results.

Consider a reaction network with $d$ species, called $\mathbf{X}_1, \ldots, \mathbf{X}_d$, and $K$ reactions. In the classical stochastic reaction network model, the dynamics is described as a continuous-time Markov chain (CTMC)[7] whose states represent the copy numbers of the $d$ network species. If the state is $x = (x_1, \ldots, x_d)$ and reaction $k$ fires, then the state is displaced by the integer stoichiometric vector $\zeta_k$. The rate of firing for reaction $k$ at state $x$ is governed by the propensity function $\lambda_k(x)$. Under the mass-action hypothesis[7]

$$\lambda_k(x_1, \ldots, x_d) = \theta_k \prod_{j=1}^{d} \frac{x_j(x_j - 1) \ldots (x_j - \nu_{jk} + 1)}{\nu_{jk}!}, \quad (2)$$

where $\theta_k$ is the rate constant and $\nu_{jk}$ is the number of molecules of $\mathbf{X}_j$ consumed by the $k$-th reaction. Formally, the CTMC $(X(t))_{t \geq 0}$ representing the reaction kinetics can be defined by its generator $\mathbb{A}$, which is an operator that specifies the rate of change of the probability distribution of the process (see Chapter 4 in ref. [38]). It is defined by

$$\mathbb{A}f(x) = \sum_{k=1}^{K} \lambda_k(x)\big(f(x + \zeta_k) - f(x)\big), \quad (3)$$

for any real-valued bounded function $f$ on the state-space which consists of all accessible states in the $d$-dimensional non-negative integer lattice.

For each state $x$, let $p(t, x)$ be the probability that the CTMC $(X(t))_{t \geq 0}$ is in state $x$ at time $t$. Then these probabilities evolve according to a system of ordinary differential equations, called the chemical master equation (CME)[7], which is typically unsolvable. Hence its solutions are often estimated with Monte Carlo simulations of the CTMC, using methods such as Gillespie's stochastic simulation algorithm (SSA)[39]. If the CME has a unique, globally attracting fixed point $\pi$ then the CTMC is called ergodic with $\pi$ as the stationary distribution. If the convergence of $p(t)$ to $\pi$ is exponentially fast in $t$, then the CTMC is called exponentially ergodic. We shall work under the assumption of exponential ergodicity which is computationally verifiable using techniques in ref. [40] and in ref. [41], wherein, it is also demonstrated that this assumption is satisfied by networks typically encountered in systems and synthetic biology. It is important to note that for an ergodic network, all stochastic trajectories, despite being different, have the same PSD.

Even though we primarily work with the CTMC model with generator (3), the PSD estimation method that we develop in this paper can also be applied to a more general CTMC model whose generator is given by

$$\mathbb{A}f(x) = \sum_{k=1}^{K} \sum_{\zeta} \lambda_k(x)\big(f(x + \zeta) - f(x)\big)\mu_k(x, \zeta), \quad (4)$$

where $\mu_k(x, \cdot)$ is a state-dependent probability distribution that governs the displacement upon firing of reaction $k$, i.e. if the state is $x$ and reaction $k$ fires, the process would jump to $(x + \zeta)$, where $\zeta$ is randomly drawn from the probability distribution $\mu_k(x, \cdot)$. Notice that by setting $\mu_k(x, \cdot)$ to be the probability distribution that puts all the mass at the fixed vector $\zeta_k$, irrespective of the state $x$, we recover the standard CTMC model with generator (3). The generality introduced by allowing the displacement to be random and state-dependent is useful in capturing cell-wide mechanisms, like cell-division, that impact the whole molecular population within a cell (see the example presented in Fig. 7).

**The resolvent operator and its connection to the PSD.** Let $(X(t))_{t \geq 0}$ be a CTMC with generator $\mathbb{A}$. For such a Markov process, we define the transition semigroup $\mathbb{T}(t)$ as the operator

which maps any real-valued function $g$ on the state space, to the function specified by the conditional expectation

$$\mathbb{T}(t)g(x) = \mathbb{E}\big(g(X(t))|X(0) = x\big). \quad (5)$$

We now define the resolvent operator which plays a central role in the development of our method for PSD estimation. For any complex number $s$, the resolvent operator maps the function $g$ to the Laplace transform of the map $t \mapsto \mathbb{T}(t)g$

$$\mathbb{R}(s)g(x) = \int_0^{\infty} e^{-st}\mathbb{T}(t)g(x)dt. \quad (6)$$

It can be shown that the map $s \mapsto \mathbb{R}(s)g(x)$ is complex-analytic.

Assuming that the observed single-cell trajectory $(X_n(t))_{t \geq 0}$ is the copy-number dynamics of the output species $\mathbf{X}_n$, we now establish a relation between the PSD $S_{X_n}(\omega)$ (see Box 1) and the resolvent operator. Let $\mathbb{E}_\pi(X_n)$ denote the stationary expectation of the copy-number of species $\mathbf{X}_n$ and let $f$ be the function

$$f(x) = x_n - \mathbb{E}_\pi(X_n). \quad (7)$$

Defining

$$G(s) := \mathbb{E}_\pi\big(f\mathbb{R}(s)f\big), \quad (8)$$

the PSD $S_{X_n}(\omega)$ is given by

$$S_{X_n}(\omega) = 2\text{Real}(G(i\omega)), \quad (9)$$

where $i = \sqrt{-1}$. This relation is proved in Section S2.2 of the Supplement. In this result we view the function $x \mapsto f(x)\mathbb{R}(s)f(x)$ as a random variable on the probability space whose sample-space is the state-space of the CTMC and the probability distribution is given by the stationary distribution $\pi$. The expectation of this random variable is denoted by $G(s)$ and in the PSD estimation method we develop, we first estimate $G(s)$ and then obtain the PSD using (9).

The eigen-decomposition of the resolvent operator allows us to express $G(s)$ as an infinite sum

$$G(s) = \sum_{j=1}^{\infty} \frac{\alpha_j}{s - \sigma_j}, \quad (10)$$

where $\sigma_1, \sigma_2, \ldots$ are the non-zero eigenvalues of $\mathbb{A}$, assumed to be distinct and arranged in descending order of their real parts (which are negative due to ergodicity). Each coefficient $\alpha_j$ captures the power in the signal corresponding to eigenmode $\sigma_j$, and their sum is equal to the total signal power which is also the stationary variance $\text{Var}_\pi(X_n)$ of the output species copy-number

$$\sum_{j=1}^{\infty} \alpha_j = \text{Var}_\pi(X_n).$$

Relation (10) is equivalent to the following representation of the autocovariance function

$$R_{X_n}(\tau) = \sum_{j=1}^{\infty} \alpha_j e^{\sigma_j \tau}. \quad (11)$$

In the case of linear networks, $G(s)$ can be exactly computed and (9) yields an analytical expression for the PSD which is already known in the literature[14]. However, for such networks stimulated by external inputs it is not known how the output PSD is related to the PSDs of the input signals. We derive this relation by exploiting the resolvent connection and this yields a practically useful PSD decomposition result (see Theorem 2.1). For general nonlinear networks, we apply the theory of Padé approximations to find an accurate rational function representation of $G(s)$ which is then used to estimate the PSD (9).

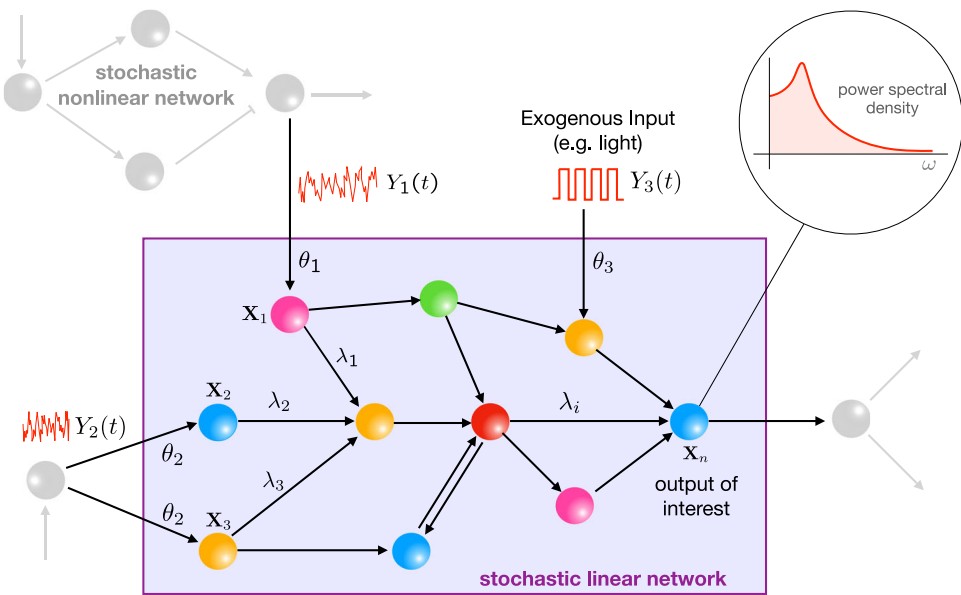

**Fig. 1 The setting of the PSD decomposition result.** A stochastic reaction network with linear propensity functions embedded in the intracellular milieu and receiving stimulation from several upstream networks. Theorem 2.1 provides an analytical decomposition for the output PSD $S_{X_n}(\omega)$ in terms of the PSDs of all the stimulating signals.

**A PSD decomposition result for linear networks**. In this section, we present a PSD decomposition result for linear networks with generator (3), that extends a similar result recently reported in ref. [26]. A reaction network is called linear if all its propensity functions are affine functions of the state variables. Under mass-action kinetics, linear networks are necessarily unimolecular, i.e. all reactions have at most one reactant and are of the form $\emptyset \longrightarrow \star$ or $X_j \longrightarrow \star$, where $\star$ represents any linear combination of species. Assuming $d$ species and $K$ reactions, for linear networks we can express the vector of propensity functions $\lambda(x) = (\lambda_1(x), \ldots, \lambda_K(x))$ as an affine map on the state-space

$$\lambda(x) = \Lambda x + \tilde{b},$$

where $\Lambda$ is some $K \times d$ matrix and $\tilde{b}$ is a $K \times 1$ vector. Letting $S$ be the $d \times K$ matrix whose columns are the stoichiometric vectors $\zeta_1, \ldots, \zeta_K$ for the reactions. We define

$$A = S\Lambda \text{ and } b = S\tilde{b},$$

and under the assumption of ergodicity, the $d \times d$ matrix $A$ is Hurwitz-stable, i.e. all its eigenvalues have strictly negative real parts. It can be easily shown (e.g. ref. [40]) that the dynamics of the expected state $x(t) = \mathbb{E}(X(t))$ is given by

$$\frac{dx}{dt} = Ax(t) + b, \tag{12}$$

and as $t \to \infty$, $x(t)$ converges to $\bar{x}$ which is the state expectation under the stationary distribution $\pi$

$$\bar{x} = \mathbb{E}_\pi(X) = -A^{-1}b.$$

Moreover, the stationary covariance matrix $\Sigma$ for the state can be computed by solving the following Lyapunov equation

$$A\Sigma + \Sigma A^T + DD^T = 0,$$

where $D$ is the positive semidefinite matrix satisfying $DD^T = S\text{diag}(\Lambda\bar{x} + \tilde{b})S^T$. In this setting, we can show that the resolvent operator maps the class of affine functions to itself, and this allows us to apply formula (9) to prove (see the Supplement, Section S2.3) that the PSD is given by

$$S_{X_n}(\omega) = -2e_n^T(\omega^2 \mathbf{I} + A^2)^{-1}A\Sigma e_n, \tag{13}$$

where $\mathbf{I}$ is the $d \times d$ identity matrix and $e_n$ denotes its $n$-th column. This expression is equivalent to the PSD formula for linear networks proved in ref. [14] using Gardiner's regression theorem[42] and it can also be derived using the LNA approximation.

Now consider the situation where such a linear network is being driven by external signals. These signals could be generated by different sources, e.g. upstream interconnected networks, environmental stimuli, or by engineered inputs introduced to probe the dynamics (see Fig. 1). A fundamentally important question is to understand how the internal noise and each of these inputs (deterministic or stochastic) conspire to make up the full power spectrum of an output of interest. Indeed it would be of considerable conceptual and practical significance to be able to decompose the output power spectrum in a way that allows the quantification of the specific contributions to the spectrum of the internal noise and of each of the external inputs. Although approximate decompositions of this sort have been reported in specific example networks modelled by CLEs[19,20], to the best of our knowledge no spectral decomposition results exist for general biochemical networks modelled by CLE, nor for those modelled by discrete stochastic CTMC models.

We consider $m$ independent time-varying signals $(Y_1(t))_{t \geq 0}, \ldots, (Y_m(t))_{t \geq 0}$. We assume that these signals stimulate through $m$ zeroth-order reactions of the form

$$\emptyset \xrightarrow{\theta_k Y_k(t)} \sum_{j=1}^{d} c_{jk}X_j \tag{14}$$

for $k = 1, \ldots, m$. Each reaction follows mass-action kinetics and for reaction $k$, $\theta_k$ is a positive constant and $c_k = (c_{1k}, \ldots, c_{dk})$ is the vector representing the number of molecules of each species $X_1, \ldots, X_d$ created by this reaction. We shall assume that process $(Y(t))_{t \geq 0}$, which includes all the stimulating signals, is an exponentially ergodic Markov process with stationary expectation $\bar{y} = (\bar{y}_1, \ldots, \bar{y}_m)$. Let $\bar{\Sigma}$ be the stationary variance-covariance matrix for the process $(X(t))_{t \geq 0}$ when each stimulating signal is deterministic and fixed to its stationary mean at all times, i.e. $Y(t) = \bar{y}$ for all $t \geq 0$. We now present our main result for linear networks which provides an analytic relationship between the

PSD $S_{X_n}(\omega)$ of our output species $\mathbf{X}_n$ and the PSDs $S_{Y_j}(\omega)$ for $j = 1, \ldots, m$.

**Theorem 2.1. (PSD Decomposition)** *Consider a linear reaction network comprising species $\mathbf{X}_1, \ldots, \mathbf{X}_d$, stimulated by independent time-varying signals $(Y_1(t))_{t \geq 0}, \ldots, (Y_m(t))_{t \geq 0}$, through zeroth-order reactions of the form* (14). *We assume that each $Y_j$ is an exponentially ergodic Markov process with PSD $S_{Y_j}(\omega)$. The PSD of the output species $\mathbf{X}_n$ is given by*

$$S_{X_n}(\omega) = \underbrace{-2e_n^T(\omega^2\mathbf{I} + A^2)^{-1}A\bar{\Sigma}e_n}_{\text{intrinsic}} + \underbrace{\sum_{j=1}^m \theta_j^2 |e_n^T(A + i\omega\mathbf{I})^{-1}c_j|^2 S_{Y_j}(\omega)}_{\text{extrinsic}}.$$

The proof of this result is provided in Section S2.3 in the Supplement and it shows that the output spectrum is the sum of the intrinsic contribution and the external contributions from all stimulating signals. The external contribution due to signal $Y_j$ is modulated by the frequency-dependent gain $\theta_j^2 |e_n^T(A + i\omega\mathbf{I})^{-1}c_j|^2$.

**Padé PSD.** In this section we develop our method, called Padé PSD, for estimating the PSD for a general nonlinear network with generator (4). For this, we apply Padé approximation theory which is known to be immensely useful in computing accurate rational function approximations for analytic functions. Recall representation (11) of the autocovariance function which is equivalent to representation (10) for the function $G(s)$. Previous studies have established that usually the autocovariance function is well-approximated by only the first few terms in this infinite series. This fact can be justified by appealing to the compactness of the resolvent operator which ensures that it is close to a finite-rank operator (see Section S2.1 in the Supplement). If we only keep the first $p$ terms in the infinite sum (10), then we obtain a rational function of the form

$$G_p(s) := \frac{\kappa_0 + \kappa_1 s + \cdots + \kappa_{p-1}s^{p-1}}{\beta_0 + \beta_1 s + \cdots + \beta_{p-1}s^{p-1} + s^p} \quad (15)$$

where the degree of the numerator polynomial is $(p-1)$ while the degree of the denominator polynomial is $p$. Based on this rational Ansatz, we shall employ the method of multipoint Padé approximation for identifying the $2p$ coefficients (viz. $\kappa_0, \ldots, \kappa_{p-1}, \beta_0, \ldots, \beta_{p-1}$) such that $G_p(s)$ serves as an accurate approximant for the function $G(s)$ given by (8), which then provides the PSD due to (9). The theory of multipoint Padé approximations[29] (also called Newton-Padé approximations[43]) is quite rich and many works have analysed their accuracy and convergence properties (see Chapter 3 in ref. [44]). In such an approximation, the rational Padé approximant is constructed by matching its power series expansions at several arbitrarily chosen points $s_1, \ldots, s_L$, up to a certain number of terms $\rho_1, \ldots, \rho_L$, to the corresponding power series expansions of the function being approximated (i.e. $G(s)$ in our case). In our application we allow each $s_\ell$ to belong to the extended positive real line $(0, \infty]$ (i.e. $\infty$ is included). The power series expansion of $G(s)$ at $s = s_\ell$ can be written as

$$G(s) = \begin{cases} a_0^{(\ell)} + a_1^{(\ell)}(s - s_\ell) + a_2^{(\ell)}(s - s_\ell)^2 + \ldots & \text{if } s_\ell < \infty \\ \frac{a_0^{(\ell)}}{s} + \frac{a_1^{(\ell)}}{s^2} + \frac{a_2^{(\ell)}}{s^3} + \ldots & \text{if } s_\ell = \infty. \end{cases} \quad (16)$$

We show in Section S2.4.1 of the Supplement that each $a_m^{(\ell)}$ can be identified as the $m$-th Padé derivative at $s = s_\ell$ defined by

$$D_m^{(s_\ell)} = \begin{cases} \frac{(-1)^m}{m!}\mathbb{E}_\pi\left(f \int_0^\infty t^m e^{-ts_\ell}\mathbb{T}(t)f dt\right) & \text{if } s_\ell < \infty \\ \mathbb{E}_\pi\left(f\mathbb{A}^m f\right) & \text{if } s_\ell = \infty, \end{cases} \quad (17)$$

where $f$ is the output function (7), $\mathbb{T}(t)$ denotes the transition semigroup operator (5) with generator $\mathbb{A}$, and $\mathbb{A}^m$ denotes the $m$-th iterate of $\mathbb{A}$ with $\mathbb{A}^0 = \mathbf{I}$ (the identity operator).

Suppose for now that these Padé derivatives have been estimated. Then it can be shown (see Section S2.4.2 in the Supplement) that for the Padé approximant $G_p(s)$ to have a power series expansion at $s = s_\ell$ that agrees with the first $\rho_\ell$ terms in (16), the $2p$-dimensional vector of unknown coefficients $x = (\kappa_0, \ldots, \kappa_{p-1}, \beta_0, \ldots, \beta_{p-1})$ must satisfy the linear system

$$A^{(\ell)}x = b^{(\ell)} \quad (18)$$

where $A^{(\ell)}$ is a $\rho_\ell \times 2p$ matrix and $b^{(\ell)}$ is a $\rho_\ell$-dimensional vector whose components in the case $s_\ell < \infty$ are given by

$$A_{ji}^{(\ell)} = \begin{cases} 0 & \text{for } i = 0, \ldots, j-1 \\ \binom{i}{j}s_\ell^{i-j} & \text{for } i = j, \ldots, p-1 \\ -\sum_{k=0}^{\min\{i-p,j\}} \binom{i-p}{k}s_\ell^{i-p-k}D_{j-k}^{(s_\ell)} & \text{for } i = p, \ldots, 2p-1 \end{cases} \quad (19)$$

and $b_j^{(\ell)} = \sum_{k=0}^j \binom{p}{k}s_\ell^{p-k}D_{j-k}^{(s_\ell)}$.

In the case $s_\ell = \infty$ these components become

$$A_{ji}^{(\ell)} = \begin{cases} 0 & \text{for } i = 0, \ldots, p-1 \text{ and } i \neq (p-1-j) \\ 1 & \text{for } i = (p-1-j) \\ 0 & \text{for } i = p, \ldots, 2p-j-1 \\ -D_{j+i-2p}^{(s_\ell)} & \text{for } i = 2p-j, \ldots, 2p-1 \end{cases}$$

and $b_j^{(\ell)} = D_j^{(s_\ell)}$.

$$\quad (20)$$

Aggregating these linear systems (18) for all $\ell = 1, \ldots, L$ we arrive at the cumulative linear system

$$Ax = b \quad (21)$$

where $A$ and $b$ are obtained by vertically stacking $A^{(\ell)}$-s and $b^{(\ell)}$-s. Note that the dimensions of $A$ and $b$ are $\rho_{\text{sum}} \times 2p$ and $\rho_{\text{sum}} \times 1$, respectively, with $\rho_{\text{sum}} = \sum_{\ell=1}^L \rho_\ell$. Hence this linear system can be underdetermined if $\rho_{\text{sum}} < 2p$ or overdetermined if $\rho_{\text{sum}} > 2p$. To handle both these possibilities in a unified way, we solve the linear system $Ax = b$ in the sense of least-squares, by minimising the residual norm $\| Ax - b\|_2^2$. This provides us with the vector of unknown coefficients $x$ to construct the rational Padé approximant $G_p(s)$.

Consider the scenario of Theorem 2.1 where the output trajectory comes from a downstream network that is driven by a stochastic external signal that emanates from an upstream network. The denominator $B(s)$ of the function $G(s)$ that characterises the PSD of the external signal can be viewed as the product of the significant eigenvalues of the generator of the upstream network (see (10)), and one can show that these are also eigenvalues for the generator of the full network that includes both the upstream and the downstream networks (see Remark S2.2 in the Supplement). Hence we can reasonably expect $B(s)$ to appear as a factor in the denominator for the function $G(s)$ that characterises the PSD of the output signal and this factor can be independently estimated from the upstream network. This suggests a more general rational Ansatz than (15), which is of the form

$$G_p(s) = \frac{\kappa_0 + \kappa_1 s + \cdots + \kappa_{p-1}s^{p-1}}{(\beta_0 + \beta_1 s + \cdots + \beta_{p-q-1}s^{p-q-1} + s^{p-q})B(s)} \quad (22)$$

where $B(s) = B_0 + B_1 s + \cdots + B_{q-1}s^{q-1} + s^q$ is some known polynomial with degree $q \leq p$. In this case, the linear system for the unknown coefficients $x = (\kappa_0, \ldots, \kappa_{p-1}, \beta_0, \ldots, \beta_{p-q-1})$ changes

from (21) to

$$A \begin{bmatrix} \mathbf{I}_p & \mathbf{0} \\ \mathbf{0} & C \end{bmatrix} x = b - A \begin{bmatrix} \mathbf{0} \\ \hat{B} \end{bmatrix} \qquad (23)$$

where $A$ and $b$ are same as before, $\mathbf{I}_p$ is the $p \times p$ identity matrix, $\hat{B} = (B_0, \dots, B_{q-1})$ is the $q$-dimensional vector of coefficients of $B(s)$ and $C$ is the $p \times (p-q)$ convolution matrix whose entries are given by

$$C_{ji} = \begin{cases} B_{j-i} & \text{if } i = j-q, j-q+1, \dots, j \\ 0 & \text{otherwise.} \end{cases}$$

For our approach to work, the main challenge is to develop a method for reliable estimation of the Padé derivatives from a handful of trajectory simulations. We describe such a method in the next section and in the subsequent sections we discuss how the resulting Padé approximant can be validated and also provide more details on the computational implementation of our Padé PSD method.

**Estimation of the Padé derivatives.** We first consider the case $s_\ell < \infty$. Appealing to the ergodicity of the CTMC we can express the Padé derivative $D_m^{(s_\ell)}$ as

$$D_m^{(s_\ell)} = \frac{(-1)^m}{s_\ell^{m+1}} \mathbb{E}_\pi \left( f \int_0^\infty \frac{t^m s_\ell^{m+1}}{m!} e^{-ts_\ell r} \mathbb{T}(t) f \, dt \right)$$
$$= \frac{(-1)^m}{s_\ell^{m+1}} \lim_{T \to \infty} \mathbb{E} \left( f(X(T)) f(X(T - \tau_{s_\ell}^{(m)})) \right) \qquad (24)$$

where $\tau_{s_\ell}^{(m)}$ is an independent random variable with `Erlang` distribution with shape parameter $(m+1)$ and rate parameter $s_\ell$. In other words, the probability density function of $\tau_{s_\ell}^{(m)}$ is given by

$$F_{\tau_{s_\ell}^{(m)}}(t) = \frac{t^m s_\ell^{m+1}}{m!} e^{-ts_\ell} \text{ for } t \geq 0,$$

and we can view $\tau_{s_\ell}^{(m)}$ as the sum of $(m+1)$ independent and identically distributed exponential random variables with rate parameter $s_\ell$. Noting that $X_n(T)$ and $X_n(T - \tau_{s_\ell}^{(m)})$ shall have the same mean and variance at stationarity we can rewrite (24) as

$$D_m^{(s_\ell)} = \frac{(-1)^m}{s_\ell^{m+1}} \left[ \text{Var}_\pi(X_n) - \frac{\delta_m^{(s_\ell)}}{2} \right], \qquad (25)$$

where $\text{Var}_\pi(X_n)$ is the stationary variance of the output species copy-number and $\delta_m^{(s_\ell)}$ is the steady-state expectation of the squared change in the output state in a time-period of length $\tau_{s_\ell}^{(m)}$, i.e.

$$\delta_m^{(s_\ell)} = \lim_{T \to \infty} \mathbb{E} \left( \left( X_n(T - \tau_{s_\ell}^{(m)}) - X_n(T) \right)^2 \right)$$

We now discuss how we can simultaneously estimate the steady-state expectation (25) for each $m = 0, 1, \dots, (\rho_\ell - 1)$. For this, we augment the CTMC state with $\rho_\ell$ additional state components, denoted by $Y_1(t), \dots, Y_{\rho_\ell}(t)$, and an extra reaction, called $\mathcal{R}_{s_\ell}$ that fires at the constant rate of $s_\ell$. If this reaction fires at time $t$, then we reset these additional state components as

$$Y_1(t) = X_n(t-) \text{ and } Y_j(t) = Y_{j-1}(t-) \text{ for } j = 2, \dots, \rho_\ell, \qquad (26)$$

where $X_n(t-)$ is the copy-number of the output species $\mathbf{X}_n$, just before the reaction firing time. Similarly for $j \geq 2$, $Y_j(t)$ assumes the value of the previous state component before the jump time, which is $Y_{j-1}(t-)$. Letting $\tau_{s_\ell}^{(m)}$ be the Erlang-distributed random

variable mentioned above, for any $T \gg 1$

$$Y_j(T) = X_n(T - \tau_{s_\ell}^{(m)}),$$

and we can express $\delta_m^{(s_\ell)}$ as

$$\delta_m^{(s_\ell)} = \lim_{T \to \infty} \mathbb{E} \left( \left( Y_{m+1}(T) - X_n(T) \right)^2 \right).$$

Suppose we have $Q$ simulated trajectories of the augmented CTMC denoted by $(X^{(q)}(t), Y^{(q)}(t))_{t \geq 0}$ for $q = 1, \dots, Q$. Then we can simultaneously estimate each $\delta_m^{(s_\ell)}$ with the Monte Carlo (MC) estimator

$$\hat{\delta}_m^{(s_\ell)} = \frac{1}{Q(T_f - T_c)} \sum_{q=1}^{Q} \int_{T_c}^{T_f} \left( Y_{m+1}^{(q)}(t) - X_n^{(q)}(t) \right)^2 dt,$$

where $T_c \ll T_f$ is the cut-off time at which stationarity is assumed to be reached and the initial part of each trajectory in the time-interval $[0, T_c]$ is discarded. Observe that if $T_f$ is large enough then even a single trajectory (i.e. $Q = 1$) is sufficient for this estimation due to Birkhoff's Ergodic Theorem[45]. However, using multiple trajectories enhances the MC estimator's statistical accuracy which can be measured by estimating its sample variance. Based on $Q$ CTMC trajectories the output variance $\text{Var}_\pi(X_n)$ can be estimated as

$$\widehat{\text{Var}}_\pi(X_n) = \frac{1}{Q(T_f - T_c)} \sum_{q=1}^{Q} \int_{T_c}^{T_f} \left( X_n^{(q)}(t) \right)^2 dt$$
$$- \left( \frac{1}{Q(T_f - T_c)} \sum_{q=1}^{Q} \int_{T_c}^{T_f} X_n^{(q)}(t) dt \right)^2. \qquad (27)$$

Plugging this estimate along with $\hat{\delta}_m^{(s_\ell)}$ in (25), we obtain estimates of the Padé derivatives $D_m^{(s_\ell)}$ for each $m = 0, \dots, (\rho_\ell - 1)$.

We now come to the case $s_\ell = \infty$. As before by simulating $Q$ CTMC trajectories we can estimate $D_m^{(\infty)}$, for each $m = 0, 1, \dots, (\rho_\ell - 1)$, using the MC estimator

$$\hat{D}_m^{(\infty)} = \frac{1}{Q(T_f - T_c)} \sum_{q=1}^{Q} \int_{T_c}^{T_f} f(X^{(q)}(t)) \mathbb{A}^m f(X^{(q)}(t)) dt. \qquad (28)$$

However, we generally find that the estimator (28) has a very large variance unless the simulation time-period $[0, T_f]$ is very large. To mitigate this issue we design suitable covariates that can be added to the integrands in (28) in order to aid convergence with respect to $T_f$ (see Section S2.4.3 in the Supplement). The resulting integrand is given by

$$\Psi_m^{(c)}(x) = -\begin{cases} \frac{1}{2}\binom{m}{r}(\mathbb{A}^r f(x))^2 + \sum_{k=1}^{r-1}\binom{m}{k}\mathbb{A}^k f(x)\mathbb{A}^{m-k}f(x) & \text{if } m = 2r \text{ is even} \\ + \sum_{k=0}^{r-1}\binom{m-1}{k}\gamma_{k(m-1-k)}(x) & \\ \sum_{k=1}^{r}\binom{m}{k}\mathbb{A}^k f(x)\mathbb{A}^{m-k}f(x) + \sum_{k=0}^{r-1}\binom{m-1}{k}\gamma_{k(m-1-k)}(x) & \text{if } m = (2r+1) \text{ is odd} \\ + \frac{1}{2}\binom{m-1}{k}\gamma_{rr}(x). & \end{cases} \qquad (29)$$

Here the function $\gamma_{jl}(x)$ is defined as

$$\gamma_{jl}(x) = \sum_{k=1}^{K} \sum_\zeta \lambda_k(x) \left[ \mathbb{A}^j(f(x+\zeta) - f(x)) \right] \left[ \mathbb{A}^l(f(x+\zeta) - f(x)) \right] \mu_k(x, \zeta). \qquad (30)$$

It can be shown that $D_m^{(\infty)} = \mathbb{E}_\pi(\Psi_m^{(c)})$ and hence we can estimate it from $Q$ CTMC trajectories as

$$D_m^{(\infty)} = \frac{1}{Q(T_f - T_c)} \sum_{q=1}^{Q} \int_{T_c}^{T_f} \Psi_m^{(c)}(X^{(q)}(t)) dt. \qquad (31)$$

In practice, we find that this covariate-based MC estimator (31) typically has much lower variance than the simpler MC estimator (28).

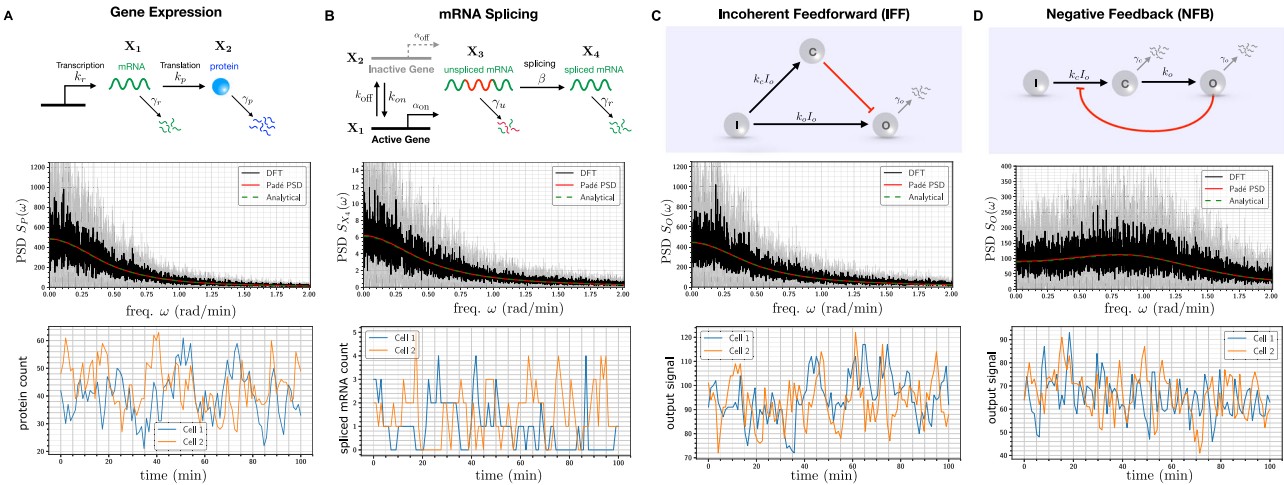

**Fig. 2 Frequency-domain analysis of linear propensity networks. A** This is the standard gene expression model where mRNA ($X_1$) is transcribed constitutively and it translates into protein ($X_2$). **B** In this RNA splicing network a gene can randomly switch between inactive ($X_2$) (low transcription) and active ($X_1$) (high transcription) states. When transcription occurs, unspliced mRNA ($X_3$) is created which is then converted into spliced mRNA ($X_4$) by the splicing machinery. **C** In the Incoherent Feedforward (IFF) network an input **I** (constant level $I_0$) directly produces the output **O** and it produces the controller species **C**, which represses the production of output **O**. **D** In the Negative Feedback (NFB) network the input **I** (constant level $I_0$) produces the controller species **C**, that produces the output species **O** which in turn inhibits the production of **C** from **I**. For all the networks single-cell output trajectories in the stationary phase are plotted. We provide a comparison of the single-cell PSDs estimated with three approaches—(1) analytically (see Table 1), (2) the Padé PSD method (see Table 1) using $Q = 10$ simulated trajectories and (3) the averaged periodogram or the DFT method mentioned in Box 1 using discrete samples from $Q = 10$ simulated trajectories. For the DFT estimator, the black curve represents the mean of the PSDs and the shaded grey region represents the symmetric one standard deviation interval around the mean. For the NFB network one can see that detecting the presence of oscillations in the fluctuations is much easier in the frequency-domain than in the time-domain.

**Validation of the Padé approximant**. Once the required Padé derivatives have been estimated, we can compute the Padé approximant $G_p(s)$ and then use this approximant to compute the PSD. For this PSD estimation procedure to work well, it is crucial that the Padé approximant $G_p(s)$ is an accurate surrogate for the function $G(s)$. This depends on many factors, such as the order of approximation $p$, the number of Padé derivatives that are estimated and their statistical precision. In order to test if a computed Padé approximant is accurate we can validate it using direct statistical estimates (i.e. without rational approximation) of the function $G(s)$ at multiple values of $s$, prescribed by $\bar{s}_1, \ldots, \bar{s}_R$. These values are all real positive numbers and similar to the Padé derivatives, the direct estimates can be estimated by augmenting the CTMC state with $R$ additional state components, denoted by $Z_1(t), \ldots, Z_R(t)$, to keep track of the copy number history of the output species $X_n$ at random exponential times in the past. Assume that there are $R$ additional reactions $\mathcal{R}_{\bar{s}_1}, \ldots, \mathcal{R}_{\bar{s}_R}$ that fire independently at constant rates $\bar{s}_1, \ldots, \bar{s}_R$, respectively. If reaction $\mathcal{R}_{\bar{s}_r}$ fires at time $t$, then we set

$$Z_r(t) = X_n(t-) \tag{32}$$

where $X_n(t-)$ is the copy-number of the output species $X_n$, just before the reaction firing time. As before we can conclude that for each $r = 1, \ldots, R$ the value $G(\bar{s}_r)$ can be estimated with $Q$ augmented CTMC trajectories, denoted by $(X^{(q)}(t), Z^{(q)}(t))_{t \geq 0}$ for $q = 1, \ldots, Q$

$$\hat{G}(\bar{s}_r) = \frac{1}{\bar{s}_r}\left[ \widehat{\mathrm{Var}}_\pi(X_n) - \frac{1}{2Q(T_f - T_c)} \sum_{q=1}^{Q} \int_{T_c}^{T_f} \left( Z_r^{(q)}(t) - X_n^{(q)}(t) \right)^2 dt \right], \tag{33}$$

where $\widehat{\mathrm{Var}}_\pi(X_n)$ is the estimator (27) for the output variance.

If the estimated Padé approximant $G_p(s)$ is accurate, each $\hat{G}(\bar{s}_r)$ would be close to the value $G_p(\bar{s}_r)$, even though both

these estimates would have some inaccuracies due to finite sampling and the finiteness of the simulation time-period. Upon comparing the graphs $\{(\bar{s}_r, \hat{G}(\bar{s}_r)) : r = 1, \ldots, R\}$ and $\{(\bar{s}_r, G_p(\bar{s}_r)) : r = 1, \ldots, R\}$, the Padé approximant can be validated.

We now present several biological examples to illustrate applications of Padé PSD method and also the PSD decomposition result for linear networks (Theorem 2.1). We start by considering some simple linear networks where analytical expressions for the exact PSDs are known and we show that Padé PSD is able to provide very accurate approximations to the PSD (see Fig. 2). Next we discuss how our PSD decomposition result allows us to identify a key criterion that enables differentiation between adapting circuit topologies[34]. We then provide two case studies to illustrate the usefulness of our PSD estimation method for synthetic biology applications. We first examine the problem of optimising the oscillation strength of the repressilator[35] (see Fig. 3) and then we consider the problem of reducing single-cell oscillations that typically arise due to the recently proposed antithetic integral feedback (AIF) controller[36] (see Fig. 4) that has the important property of ensuring robust perfect adaptation for arbitrary intracellular networks with stochastic dynamics. Next, we examine how the PSD decomposition result can help us in studying the phenomenon of single-cell entrainment in the stochastic setting (see Fig. 5) and then we present an example to show how Padé PSD facilitates parameter inference with experimental single-cell trajectories that measure the copy-numbers of the output species up to an unknown constant of proportionality (see Fig. 6). Lastly, we consider an example with cell-division cycle, and demonstrate that our Padé PSD method can be used for accurately estimating the PSDs and quantitatively examining oscillations induced by the cell-cycle (see Fig. 7).

Detailed descriptions of the networks considered in the paper and their PSD analysis can be found in Section S4 of the

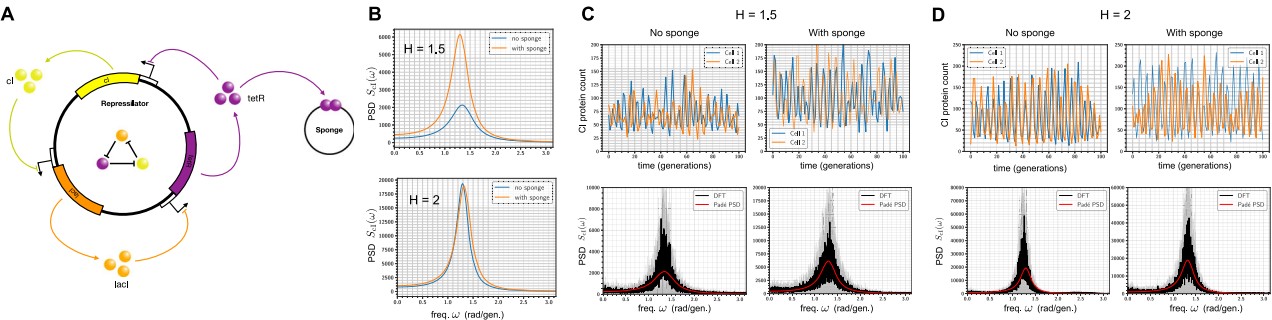

**Fig. 3 Improving the repressilator's oscillatory strength. A** Depiction of the repressilator network with three gene expression systems whose output proteins cyclically repress each other. When present, the sponge plasmid can bind TetR proteins, thereby raising the derepression threshold of the *cl* gene. **B** Shows the effect of the sponge on the PSD. It can be seen that the sponge sharpens the PSD peak for promoter cooperativity $H = 1.5$ but this effect is lost for $H = 2$. **C** Plots the single-cell trajectories with and without the sponge for promoter cooperativity $H = 1.5$, and they show that the oscillations are more regular in the latter case. Comparison of the PSD estimated with our Padé PSD method with the PSDs estimated with DFT is provided. For these estimations, $Q = 10$ simulated trajectories were used. **D** Repeats the computational analysis in panel (**C**) for promoter cooperativity $H = 2$.

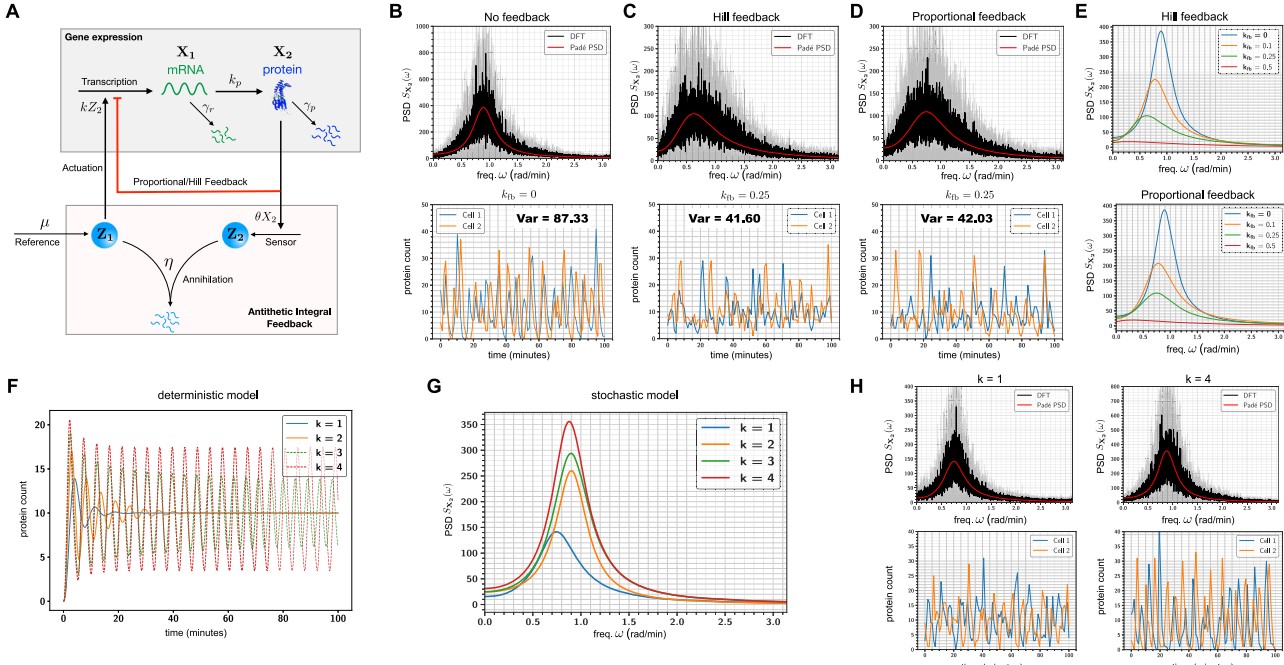

**Fig. 4 Reducing single-cell oscillations due to the AIF controller. A** Depiction of the biomolecular antithetic integral feedback (AIF) controller regulating the gene expression network. Here mRNA ($\mathbf{X_1}$) is the actuated species and the protein ($\mathbf{X_2}$) is the output species. This protein output is sensed by the controller species $\mathbf{Z_2}$ which annihilates the other controller species $\mathbf{Z_1}$ that is constitutively produced at rate $\mu$. The species $\mathbf{Z_1}$ actuates the gene expression network by catalysing the production of mRNA $\mathbf{X_1}$. The red arrow indicates an extra negative feedback from the output species (protein) to the production of the actuated species (mRNA). In **B**, the single-cell oscillatory trajectories for the protein counts (without the extra feedback) are plotted and the corresponding PSD is estimated with Padé PSD and the DFT method. **C** Same plots as in panel (**B**) for Hill feedback with $k_{fb} = 0.25 \, \text{min}^{-1}$. **D** Same plots as in panel (**B**) for proportional feedback with $k_{fb} = 0.25 \, \text{min}^{-1}$. For other values of $k_{fb}$, comparison plots between Padé PSD and DFT are provided in Fig. S5(A) in the Supplement. The plots for the single-cell trajectories in panels (**B**–**D**) also indicate the total signal power which is equal to the stationary output variance (see Box 1). Notice the $\geq 50\%$ reduction in this variance in the presence of feedback. **E** Comparison of the PSDs estimated with the Padé PSD method for the Hill and proportional feedback for three choices of feedback parameter $k_{fb}$. In **F**, we simulate the deterministic model for this network (without the extra feedback) for four values of the actuation rate constant $k$. Notice that for lower values of $k$, the deterministic trajectories converge to a fixed point, but for higher values of $k$ sustained oscillations emerge. This is quite different from the stochastic case, where oscillations persist even though their power continuously decreases as k decreases, as seen in panel (**G**) from the plots of PSDs obtained with the Padé PSD method. **H** Comparison of the PSDs estimated with Padé PSD and the DFT method for two values of $k$ (for other values see Fig. S5(B) in the Supplement). All the PSDs were estimated with $Q = 10$ simulated trajectories.

Supplement. Unless otherwise stated, all reaction networks are assumed to follow CTMC dynamics with generator (3) and all propensity functions are assumed to be of the mass-action form (2).

**Validation of Padé PSD with linear networks**. We now provide analytical expressions for the PSD of certain simple networks, like the birth-death, the classical gene expression network[46] and the recently proposed RNA splicing network[47]. We then show that Padé PSD is able to approximate the PSD quite accurately.

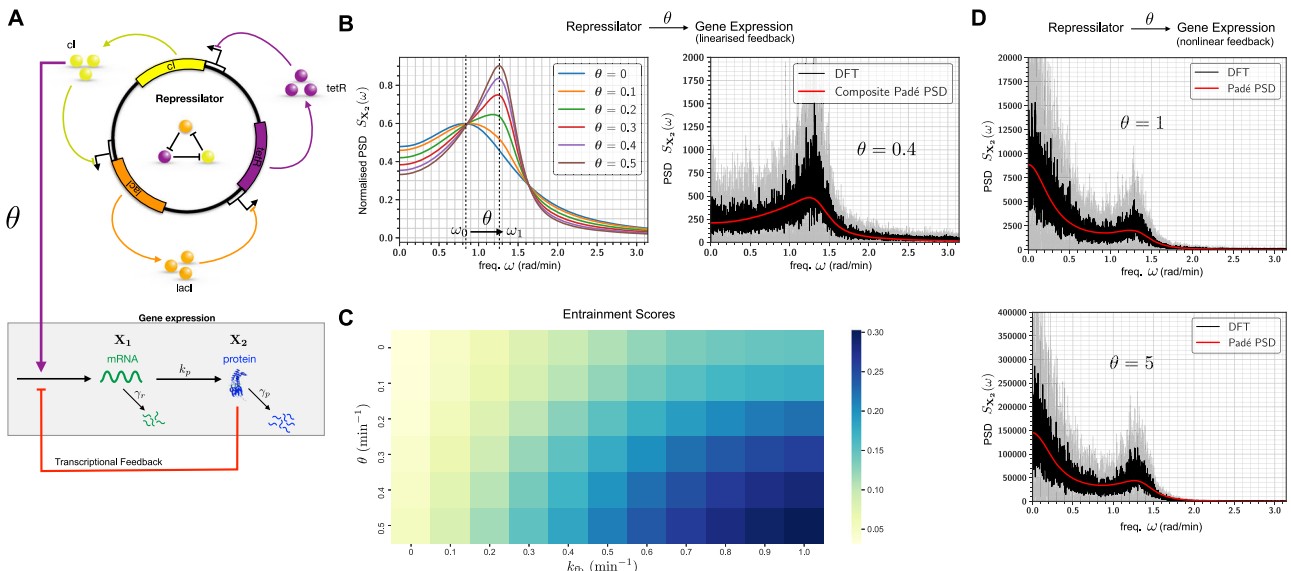

**Fig. 5 Stochastic entrainment of gene expression by the repressilator.** **A** Schematic diagram of the repressilator driving a gene expression network. The cI protein from the repressilator acts as an activating transcription factor for mRNA ($\mathbf{X}_1$) which translates into output protein ($\mathbf{X}_2$). The red arrow from $\mathbf{X}_2$ to $\mathbf{X}_1$ indicates negative transcriptional feedback from the protein molecules. When the repressilator is connected to the gene expression network, for linearised feedback the PSD can be estimated with the composite Padé PSD method which is based on Theorem 2.1. In **B**, these PSD estimates (after normalisation by the total area under the PSD curve) are plotted for six values of $\theta$ and compared for $\theta = 0.4\,\mathrm{min}^{-1}$ to the PSD obtained with the DFT method. One can observe the stochastic entrainment phenomenon as $\theta$ increases. **C** The heat-map for the entrainment score (see (44)) as a function of $\theta$ and the feedback strength parameter $k_{\mathrm{fb}}$. Observe that the entrainment score is monotonically increasing in both variables $k_{\mathrm{fb}}$ and $\theta$, but it is more sensitive to $k_{\mathrm{fb}}$. **D** For nonlinear transcriptional feedback PSD estimates obtained with Padé PSD are plotted and compared with the DFT method for $\theta = 1\,\mathrm{min}^{-1}$ and $\theta = 5\,\mathrm{min}^{-1}$. All the PSDs were estimated with $Q = 10$ simulated trajectories.

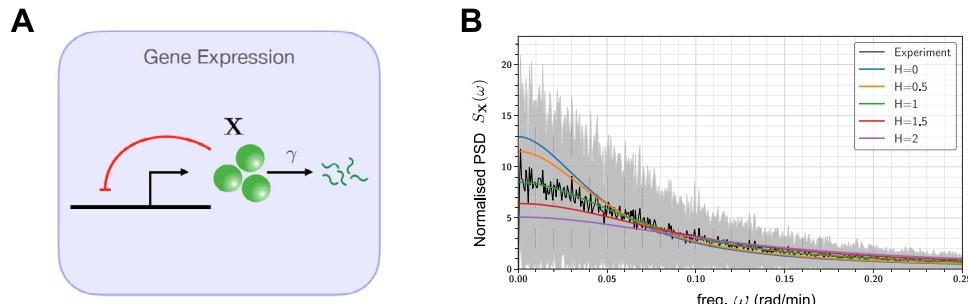

**Fig. 6 PSD-based inference of a self-regulatory gene expression model.** **A** Depicts the self-regulatory gene expression system where the output represses the gene (shown in red) via a nonlinear Hill function (45) with cooperativity coefficient $H$. **B** Plots the normalised PSDs (area under the PSD curve is 1) obtained by Padé PSD for various values of $H$, and compares it with the normalised PSD obtained from experimental single-cell trajectories. The experimental PSD was computed by averaging the PSDs from $Q = 100$ single-cell trajectories, and the black curve represents the mean, while the shaded grey region represents the symmetric one standard deviation interval around the mean.

*Gene transcription.* Consider a simple model of constitutive gene transcription and mRNA degradation, given by a single-species birth-death network with rate of production $k$ and the rate of degradation $\gamma$

$$\emptyset \xrightarrow{k} \mathbf{X} \xrightarrow{\gamma} \emptyset.$$

The stationary distribution for this network is Poisson with parameter $k/\gamma$. Hence the stationary mean and variance and equal to $k/\gamma$ and applying formula (13) we can compute the PSD as

$$S_X(\omega) = \frac{2k}{\gamma^2 + \omega^2}. \tag{34}$$

This shows that the PSD (normalised by the total area under its curve) has the fat-tailed Cauchy distribution with infinite mean and variance, showing that even for such a simple network the stochastic output trajectory contains a very wide range of frequencies.

*Gene expression network.* We now analyse the gene expression model shown in Fig. 2A that consists of two species—the mRNA ($\mathbf{X}_1$) and the protein ($\mathbf{X}_2$). There are four reactions corresponding to mRNA transcription, protein translation and the first-order degradation of both the species. Observe that the mRNA dynamics is birth-death and hence we can compute its PSD using (34) with $(k, \gamma) \mapsto (k_r, \gamma_r)$. Since mRNA stimulates the creation of protein via a reaction of the form (14) we can apply our PSD decomposition result (Theorem 2.1) to express the protein PSD as a sum of two components corresponding to translation and transcription, respectively:

$$S_{X_2}(\omega) = \underbrace{\frac{2k_r k_p}{\gamma_r(\gamma_p^2 + \omega^2)}}_{\text{translation}} + \underbrace{\frac{k_p^2}{\gamma_p^2 + \omega^2}\frac{2k_r}{\gamma_r^2 + \omega^2}}_{\text{transcription}}. \tag{35}$$

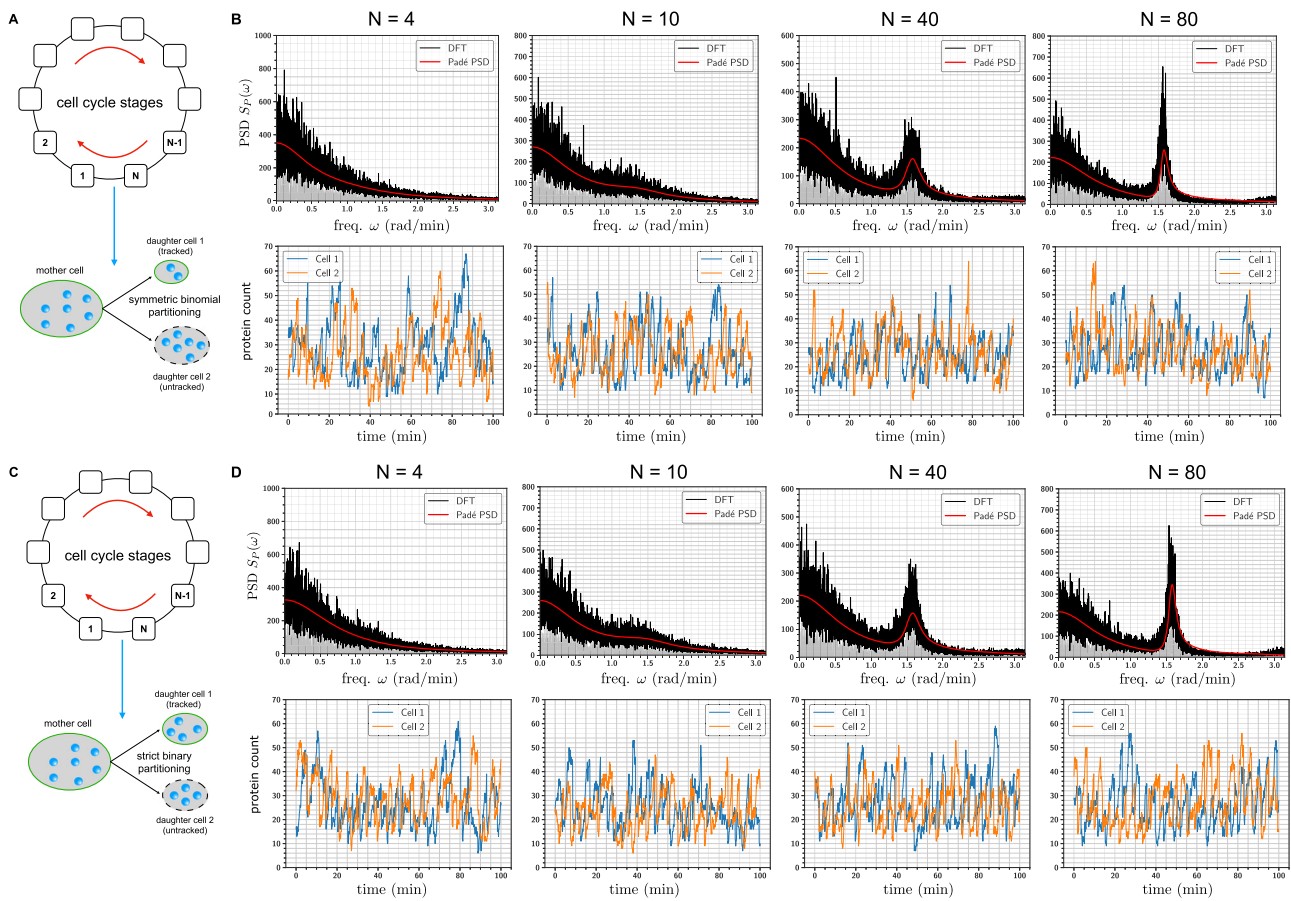

**Fig. 7 Cell-cycle induced oscillation in gene expression.** We model the cell-cycle evolution as a $N$-stage Markov process with a constant rate $\alpha$ of transitioning from one stage in the cycle to the next. When the transition is from state $N$ to state 1, the mother cell splits into two daughter cells, from which only one cell is tracked and measured. At the time of the split, the mother cell molecules are partitioned in one of two ways—symmetric binomial (i.e. each molecule has a 50% chance of ending up in the tracked cell) and strict binary (i.e. the tracked cell receives exactly half of the molecules of each network species). These two scenarios are depicted in panels (**A**) and (**C**). In panel **B** (resp. panel (**D**)), we suppose that the cell undergoing symmetric binomial (resp. strict binary) partitioning, contains the gene expression network shown in Fig. 2A. We plot the single-cell trajectories for the protein counts as well as the corresponding PSDs estimated with the Padé PSD and the DFT methods for four values of the cell-cycle length $N$, keeping the cell-cycle frequency constant. All the PSDs were estimated with $Q = 10$ simulated trajectories.

The translation term is computed by setting the mRNA level to its stationary mean $\bar{x}_1 := k_r/\gamma_r$ and then viewing the protein dynamics as a birth-death process with production rate $k_p\bar{x}_1$ and degradation rate $\gamma_p$. The transcription term is simply the PSD of mRNA modulated by the frequency-dependent factor given by Theorem 2.1.

*RNA splicing network.* The recently proposed RNA Splicing network (see Fig. 2B) was used to model the concept of RNA velocity that can help in understanding cellular differentiation from single-cell RNA-sequencing data[47]. Here a single gene-transcript can randomly switch between active ($\mathbf{X}_1$) and inactive ($\mathbf{X}_2$) states with different rates of transcription of unspliced mRNA ($\mathbf{X}_3$). The splicing process converts these unspliced mRNAs into spliced mRNAs ($\mathbf{X}_4$). Both spliced and unspliced mRNAs undergo first-order degradation. Applying formula (13) we can write the PSD of the dynamics of active gene count as

$$S_{X_1}(\omega) = \frac{2k_{on}k_{off}}{(k_{on} + k_{off})((k_{on} + k_{off})^2 + \omega^2)}. \quad (36)$$

Note that when the active gene count is $X_1 \in \{0, 1\}$ the transcription rate is $\alpha_{off} + (\alpha_{on} - \alpha_{off})X_1$. We can view transcription as a superposition of two reactions—a constitutive reaction with

rate $\alpha_{off}$ and reaction of the form (14) where the stimulant is the active gene $\mathbf{X}_1$. Applying Theorem 2.1 we can decompose the PSD of the spliced mRNA count as

$$S_{X_4}(\omega) = \underbrace{\frac{2\beta(\alpha_{off}k_{off} + \alpha_{on}k_{on})}{(\beta + \gamma_u)(k_{off} + k_{on})(\gamma_r^2 + \omega^2)}}_{\text{splicing}} + \underbrace{\frac{\beta^2(\alpha_{on} - \alpha_{off})^2}{((\beta + \gamma_u)^2 + \omega^2)(\gamma_r^2 + \omega^2)}S_{X_1}(\omega)}_{\text{transcription}},$$

(37)

where $S_{X_1}(\omega)$ is given by (36).

Observe that for both gene expression and RNA splicing networks we can find an analytical expression for the PSD by directly applying formula (13) for the full network. However, using our PSD decomposition result we not only simplify the computation but also identify the contribution of the network mechanisms to the PSD.

For a specific parameterisation of these two networks, we compare the PSDs obtained analytically with those obtained by our Padé PSD method and the standard periodogram estimator for PSD that is based on discrete-sampling and DFT (see Box 1). The results are presented in Fig. 2A, B and they show good agreement, despite the noisy nature of the DFT estimate. The analytical expressions for the PSD along with the PSD estimates produced by Padé PSD are given in Table 1. One can see that the

**Table 1 Expressions for PSDs estimated analytically and with the Padé PSD method.**

| Network | Analytical PSD | Padé PSD |
|---|---|---|
| Gene expression | $\dfrac{40\omega^2+120}{\omega^4+1.25\omega^2+0.25}$ | $\dfrac{39.9567\omega^2+118.3381}{\omega^4+1.2328\omega^2+0.2442}$ |
| RNA splicing | $\dfrac{1.2\omega^4+30\omega^2+219.84}{\omega^6+25.25\omega^4+150.25\omega^2+36}$ | $\dfrac{1.2010\omega^4+6.9450\omega^2+9.4755}{\omega^6+5.2220\omega^4+7.3450\omega^2+1.5276}$ |
| IFF | $\dfrac{94\omega^2+112}{\omega^4+1.25\omega^2+0.25}$ | $\dfrac{93.988\omega^2+114.53}{\omega^4+1.2752\omega^2+0.2527}$ |
| NFB | $\dfrac{66.6667\omega^2+200}{\omega^4-0.75\omega^2+2.25}$ | $\dfrac{66.6301\omega^2+201.8281}{\omega^4-0.7166\omega^2+2.2159}$ |

To estimate the Padé approximant we use $\mathbf{s} = (1, 1.5, 2, \infty)$ and $\boldsymbol{\rho} = (2, 2, 2, 3)$ for the RNA splicing network and for all other networks we use $\mathbf{s} = (\infty)$ and $\boldsymbol{\rho} = (4)$.

PSD estimated by our method is quite "close" to the analytical PSD for the gene expression network. The same holds for the RNA splicing network (see the PSD plots in Fig. 2B) even though it is not apparent from the expressions in Table 1.

**The PSD enables discrimination between regulatory topologies**. We consider simple three-node IFF and NFB topologies depicted in Fig. 2C, D with stochastic kinetics. We provide analytical expressions for the PSDs under the assumption of linearised propensity functions for the repression mechanisms. These expressions inform us about qualitative structural differences between the PSDs obtained from IFF and NFB topologies, regardless of the choice of reaction rate parameters. This shows that in the stochastic setting, the PSD of single-cell trajectories serves as a key "response signature" that can differentiate between adapting circuit topologies. We demonstrate this finding with our Padé PSD model for a specific parametrisation of these networks and we argue why this result holds for arbitrarily-sized IFF and NFB networks.

We begin by analysing the IFF topology, where the controller species **C** catalytically produces the output species **O** at rate $F_f(x_c)$ which is a monotonically decreasing function of the controller species copy-number $x_c$ and it represents the repression of **O** by **C**. We linearise the function $F_f(x_c)$ as

$$F_f(x_c) = \beta_0 - \beta_{ff}x_c, \tag{38}$$

where $\beta_0$ and $\beta_{ff}$ are positive constants denoting the basal production rate and the strength of the incoherent feedforward mechanism, respectively. With this linearisation, all propensity functions become affine and hence we can apply the results for linear networks. Specifically, the steady-state means $\bar{x}_c := \mathbb{E}_\pi(C)$ and $\bar{x}_o := \mathbb{E}_\pi(O)$ are given by

$$\bar{x}_c = \frac{k_c I_0}{\gamma_c} \text{ and } \bar{x}_o = \frac{k_o I_0 + \beta_0}{\gamma_o} - \frac{\beta_{ff}k_c I_0}{\gamma_c \gamma_o}$$

and it is immediate that if $\beta_{ff} \approx k_o\gamma_c/k_c$, then the mean output value $\bar{x}_o \approx \beta_0/\gamma_o$ becomes insensitive to the input abundance level $I_0$. This shows the adaptation property of the IFF network.

As the dynamics of **C** is simply birth-death with production rate $k_c I_0$ and degradation rate $\gamma_c$, its PSD is given by

$$S_C(\omega) = \frac{2k_c I_0}{\gamma_c^2 + \omega^2}.$$

Under the assumption of linearity of the feedforward function $F_f$ the stimulation of **O** by **C** can be viewed as zeroth-order degradation. Applying Theorem 2.1 we can evaluate the output PSD as

$$S_O(\omega) = \frac{2(k_o I_0 + \beta_0 - \beta_{ff}\bar{x}_c)}{\gamma_o^2 + \omega^2} + \frac{\beta_{ff}^2}{\gamma_o^2 + \omega^2}S_C(\omega).$$

Since this is a sum of two non-negative monotonically decreasing functions of $\omega$, we can conclude that $S_O(\omega)$ is also

monotonically decreasing. Hence output trajectories cannot show oscillations regardless of the IFF network parameters. This same argument can be extended to IFF networks with arbitrary number of nodes (see the Supplement, Section S4.1.3).

In the NFB topology, the production of the controller species **C** is repressed by the output species **O**, and we model the production rate by a monotonically decreasing function $F_b(x_o)$ of the output species copy-number $x_o$. As before, we linearise this function as

$$F_b(x_o) = \beta_0 - \beta_{fb}x_o, \tag{39}$$

where $\beta_0$ is the basal production rate and $\beta_{fb}$ is the feedback strength. Under this linearisation, the steady-state means $\bar{x}_c := \mathbb{E}_\pi(C)$ and $\bar{x}_o := \mathbb{E}_\pi(O)$ are given by

$$\bar{x}_c = \frac{\gamma_o\beta_0 I_0}{\gamma_c\gamma_o + k_o\beta_{fb}I_0} \text{ and } \bar{x}_o = \frac{k_o\beta_0 I_0}{\gamma_c\gamma_o + k_o\beta_{fb}I_0}.$$

Observe that if the input abundance level $I_0$ is high, then mean output value $\bar{x}_o \approx \beta_0/\beta_{fb}$ only depends on the feedback function $F_b$ and it is insensitive to $I_0$, thereby demonstrating the adaptation property. Applying formula (13) we arrive at the following expression for the PSD for the output trajectory

$$S_O(\omega) = \frac{2\gamma_o k_o\beta_0 I_0}{\gamma_c\gamma_o + k_o\beta_{fb}I_0}\left[\frac{\gamma_c^2 + k_o\gamma_c + \omega^2}{(\gamma_c\gamma_o + k_o\beta_{fb}I_0)^2 + \omega^2(\gamma_c^2 + \gamma_o^2 - 2k_o\beta_{fb}I_0) + \omega^4}\right]. \tag{40}$$

Proposition S4.1 in the Supplement proves that the mapping $\omega \mapsto S_O(\omega)$ has a positive local maximum (which is also the global maximum) if and only if

$$k_o\beta_{fb}I_0 > \frac{\gamma_c^4 + \gamma_c^3 k_o + \gamma_c^2\gamma_c k_o}{\sqrt{\Gamma(\gamma_c, \gamma_o, k_o)} + \gamma_c\gamma_o + \gamma_c^2 + k_o\gamma_c}, \tag{41}$$

where $\Gamma(\gamma_c, \gamma_o, k_o) := (\gamma_c\gamma_o + \gamma_c^2 + k_o\gamma_c)^2 + \gamma_c^4 + \gamma_c^3 k_o + \gamma_o^2\gamma_c k_o$. This condition shows that regardless of the choice of NFB network parameters, the output trajectories will exhibit oscillation if the input abundance level $I_0$ is high enough. Using the standard root-locus argument[48] we can draw the same conclusion for arbitrarily-sized NFB networks (see the Supplement, Section S4.1.3). This shows that the existence of oscillations and non-monotonicity of the PSD is a differentiator between the NFB and the IFF networks as the latter never exhibits oscillations. Note that high $I_0$ is precisely the condition for NFB to show adaptation and hence imposing this requirement is not very restrictive. The role of negative feedback in causing stable stochastic oscillations was explored theoretically in ref. [27] with CLE, and it has also been demonstrated experimentally.

For a specific parameterisation of the three-node IFF and NFB networks, we compare the PSD produced by our method with the analytical PSD and the DFT-based estimator. The results are shown in Fig. 2C, D and one can see that Padé PSD is quite accurate in estimating the PSD, which is also evident from the PSD expressions provided in Table 1. Since negative propensities cannot be allowed, we perform simulations with the positive part of the linear feedforward (see (38)) and feedback (see (39)) functions. Hence the analytical PSD expressions are not exact but they are still close because the dynamics rarely enters the states for which these linear functions become negative.

**Using the PSD for enhanced oscillator design**. The repressilator[35] is the first synthetic genetic oscillator and it consists of three genes repressing each other in a cyclic fashion (see Fig. 3A). These three genes are *tetR* from the Tn10 transposon, *cI* from bacteriophage $\lambda$ and *lacI* from the lactose operon. These three genes create three repressor proteins which are TetR, cI and

LacI, respectively, and the cyclic repression mechanism can be represented as

$$\text{TetR} \dashv \text{cI} \dashv \text{LacI} \dashv \text{TetR}.$$

Due to intrinsic noise in the dynamics, the repressilator loses oscillations at the bulk or the population-average level after a few generations. At the single-cell level this intrinsic noise broadens the output PSD peak, making the oscillations less regular in both amplitude and phase. In other words, intrinsic noise compromises the ability of the circuit to keep track of time. This issue was addressed in a recent paper[49] which elaborately studied the various sources of noise in the original circuit and eliminated them to construct a modified repressilator circuit that showed regular oscillations over several generations. It was found that most of the noise was generated when TetR protein levels were low and the derepression of the TetR controlled promoter occurred at a low threshold. To raise this threshold a sponge plasmid was introduced and this had the remarkable effect of regularising the oscillations and sharpening the single-cell PSD peak.

It is also known that increasing the cooperativity of the repression mechanism improves regularity of the oscillations[35]. A fundamental question then arises is that—does the PSD-sharpening effect of the sponge plasmid persist when the repression cooperativity is increased? If this is true then one can regularise oscillations even more by designing cooperative promoters in addition to employing the sponge device. We study this question using an adaptation of the stochastic model given in ref. [49]. The stochastic model is detailed in Section S4.2.1 of the Supplement. The repression mechanism is encoded with a nonlinear Hill function whose coefficient $H$ represents the degree of cooperativity among the promoter binding sites. The sponge plasmid, if present, can competitively bind the free TetR molecules, reducing the number of these molecules available for repressing the $cI$ gene.

We demonstrate that our method is able to accurately estimate the single-cell PSD and exhibit the sharpening of the PSD in the presence of the sponge plasmid when the cooperativity is set to $H = 1.5$. Surprisingly when the cooperativity is increased to $H = 2$, the sponge loses its effect of sharpening the PSD. This shows that in certain parameter regimes, the oscillation-regularising effects of the sponge plasmid and the repressor binding cooperativity are not additive, possibly due to the fact that increased cooperativity makes the repression mechanism more ultrasensitive[50].

With our method, we estimate the PSD for the dynamics of the copy-numbers of the cI protein, whose expression is directly repressed by TetR. For the promoter cooperativity (i.e. the Hill coefficient) of $H = 1.5$, the PSD indeed exhibits a sharper peak, in the presence of the sponge plasmid, at the peak frequency of around $\omega_{\max} \approx 1.35$ rad./gen. (see Fig. 3B). This sharpness in PSD suggests more regularity in oscillations which is also evident from the single-cell trajectories plotted in Fig. 3C. We compare our PSD estimation method with the DFT method in both the cases (with and without sponge) and the results are shown in Fig. 3C. The same analysis is repeated for the promoter cooperativity of $H = 2$ and the results are shown in Fig. 3B and D. From Fig. 3B it is immediate that for $H = 2$, the PSD sharpening effect of the sponge plasmid is lost.

**Biocontroller design with PSD: suppressing single-cell oscillations.** In recent years genetic engineering has allowed researchers to implement biomolecular control systems within living cells (see refs. [36,51–60]). This area of research, popularly known as *Cybergenetics*[51], offers promise in enabling control of living cells

for applications in biotechnology[61,62] and therapeutics[63]. A particularly important challenge in Cybergenetics is to engineer an intracellular controller that facilitates cellular homoeostasis by achieving robust perfect adaptation (RPA) for an output state-variable in an arbitrary intracellular stochastic reaction network. This challenge was theoretically addressed in ref. [36] which introduced the antithetic integral feedback (AIF) controller and demonstrated its ability to achieve RPA for the population-mean of output species. This controller has been synthetically implemented in vivo in bacterial cells, and it has been shown that any biomolecular controller that achieves RPA for arbitrary reaction networks with noisy dynamics, must embed this controller[60].

Computational analysis has revealed that AIF controller can cause high-amplitude oscillations in the single-cell dynamics in certain parameter regimes[36,64] which could potentially be undesirable and/or unfavourable. Hence it is important to find ways to augment the AIF controller, so that single-cell oscillations are attenuated but the RPA property is preserved. It is known that adding an extra negative feedback (like proportional action) from the output species to the actuated species maintains the RPA property, while decreasing both the output variance and the settling-time for the mean dynamics[65]. Using the PSD estimation method developed in this paper we now demonstrate how adding such a negative feedback also helps in diminishing single-cell oscillations.

The AIF controller is depicted in Fig. 4A and it is acting on the gene expression model considered in Fig. 2A. The AIF controller robustly steers the mean copy-number level of the protein $X_2$ to the desired set-point $\mu/\theta$, where $\mu$ is the production rate of $Z_1$ and $\theta$ is the reaction rate constant for the output sensing reaction. The AIF affects the output by actuating the production of mRNA $X_1$ and the feedback loop is closed by the annihilation reaction between $Z_1$ and $Z_2$. This annihilation reaction can be viewed as mutual inactivation or sequestration and it can be realised using biomolecular pairs such as sigma/anti-sigma factors[54,66,67], scaffold/anti-scaffold proteins[68] or toxin/antitoxin proteins[69].

It is known from ref. [36] that the combined closed-loop dynamics is ergodic and mean steady-state protein copy-number is $\mu/\theta$

$$\lim_{t \to \infty} \mathbb{E}(X_2(t)) = \frac{\mu}{\theta}.$$

As discussed in ref. [65], this ergodicity is preserved under certain conditions when an extra negative feedback from protein $X_2$ to the production of mRNA $X_1$ is added. Letting $z_1$ and $x_2$ denote the copy-numbers of $Z_1$ and $X_2$, respectively, we add the extra feedback by changing the rate of the actuation reaction from $k z_1$ to $(k z_1 + F_b(x_2))$ where $F_b$ is a monotonically decreasing feedback function which takes non-negative values. As in ref. [65], we consider two types of feedback. Letting $\hat{\mu}$ to be the reference point, the first is Hill feedback of the form

$$F_b(x_2) = \frac{4 k_{\text{fb}} \hat{\mu}^2}{\hat{\mu} + x_2}$$

which is based on the actual output copy-number $x_2$, while the second is the proportional feedback that is essentially the linearisation of the Hill feedback at the reference point $\hat{\mu}$

$$F_b(x_2) = k_{\text{fb}} \max\{3\hat{\mu} - x_2, 0\}.$$

One can easily see that at the reference point, the values of this feedback function $F_b(\hat{\mu})$ and its derivative $F_b'(\hat{\mu})$ (equal to $-k_{\text{fb}}$) are the same for both types of feedback. We can view $k_{\text{fb}}$ as the feedback gain parameter. The Hill feedback is biologically more realisable, while the proportional feedback captures the classical controller where the feedback strength depends linearly on the deviation of the output $x_2$ from the reference point $\hat{\mu}$, in the

output range $[0, 3\hat{\mu}]$. In our analysis, we set the reference point $\hat{\mu}$ as the set-point $\mu/\theta$.

For a particular network parametrization, we use our method to estimate the PSD for the single-cell protein dynamics in the AIF-regulated gene expression network, and the results are displayed in Fig. 4. When the extra negative feedback is absent (i.e. $k_{fb} = 0$) the single-cell trajectory has high-amplitude oscillations which is also evident from the estimated PSD (see Fig. 4B). In Fig. 4C, we apply our Padé PSD method to examine how the PSD changes when extra feedback of Hill type is added with varying strengths given by parameter $k_{fb}$. Observe that as the feedback strength increases, the PSD peak declines and the oscillations become almost non-existent for $k_{fb} = 0.5 \text{ min}^{-1}$. The same holds true for the proportional feedback (see Fig. 4D). These results suggest that both feedback mechanisms are more or less equally effective in reducing oscillations. This is further corroborated by the single-cell trajectories plotted in Fig. 4C, D which also shows that addition of feedback decreases the stationary output variance, that is equal to the signal power (see Box 1). In a recent paper, the decrease in oscillations upon addition of proportional feedback has been experimentally validated in yeast cells[70].

In ref. [36], it is reported that the deterministic model of the AIF-regulated gene expression network can exhibit both convergence to a fixed point and sustained oscillations. Keeping all other parameters fixed and setting $k_{fb} = 0$, we simulate the deterministic model for four values of the actuation rate constant $k$ and plot the output protein trajectories in Fig. 4F. One can see that for lower values of $k$, the deterministic trajectories converge to a fixed point, which is equal to the set-point $\mu/\theta$, while for higher values of $k$, the trajectories oscillate around the set-point. Estimating the PSDs for the stochastic model with our Padé PSD method we find that for all the four $k$ values, the PSDs have a non-zero peak around 1 rad/ min (see Fig. 4G). This shows that the oscillatory tendency of the stochastic model persists, albeit at lower PSD peak values, for values of $k$ beyond the critical value where the deterministic system transitions from a limit cycle to a fixed point. For the lower values of $k$, oscillations are noise-induced in the sense that they only emerge in the presence of randomness in the dynamics and they disappear (at steady state) if the noise-free deterministic model is considered. For two values of $k$, we plot the PSD obtained by our method and compare it with the PSD estimated with DFT and one can see from Fig. 4H that there is good agreement. The details on all the computations for the AIF-regulated gene expression network can be found in Section S4.2.2 of the Supplement.

This example with noise-induced oscillations also shows that the LNA would yield a very inaccurate PSD estimate as it essentially adds a Gaussian term to the deterministic dynamics. Hence if the deterministic dynamics converge to a fixed point, the LNA-based PSD estimator cannot have a peak at a non-zero frequency value.

**Exploiting the PSD for studying stochastic entrainment**. The phenomenon of entrainment occurs when an oscillator, upon stimulation by a periodic input, loses its natural frequency and adopts the frequency of the input. This phenomenon has several applications in physical, engineering and biological systems[71]. The most well-known biological example of this phenomenon is the entrainment of the circadian clock oscillator by day-night cycles. The circadian clock is an organism's time-keeping device and its entrainment is necessary to robustly maintain its periodic rhythm[72]. The circadian clock is one example among several intracellular oscillators that have been found and their functional roles have been identified[73]. Often these oscillators provide entrainment cues to other networks within cells[74] and hence it is

important to study entrainment at the single-cell level, where the dynamics is intrinsically noisy due to low copy-number effects.

We now illustrate how our PSD decomposition result (Theorem 2.1) can be used to study single-cell entrainment in the stochastic setting where the dynamics is described by CTMCs. We consider the example of the repressilator stimulating a gene expression system, as shown in Fig. 5A. This gene expression network is the same as in Fig. 2A but we include transcriptional feedback from the protein molecules and so the mRNA transcription rate is given by a monotonic decreasing function $F_b(x_2)$ of the protein copy-number $x_2$. We shall linearise $F_b(x_2)$ as

$$F_b(x_2) = k_r - k_{fb}x_2,$$

where $k_r$ is the basal transcription rate and $k_{fb}$ is the feedback strength. When this gene expression network is connected to the repressilator (see Fig. 5) the transcription rate changes from $F_b(x_2)$ to

$$\theta p_2 + F_b(x_2), \tag{42}$$

where $p_2$ is the molecular count of protein cI in the repressilator and parameter $\theta$ captures the "strength" of the interconnection. In other words, cI acts as an activating transcription factor in our example. The parameters of the repressilator are chosen as in Fig. 3 in the "no sponge" and Hill coefficient $H = 1.5$ case, but the time-units are changed to minutes. We can view the gene expression network as simply the negative feedback (NFB) network in Fig. 2 with the controller species **C** as mRNA $\mathbf{X}_1$ and the output species **O** as protein $\mathbf{X}_2$. Using the same parameters as the NFB network, we study how the PSD of the protein output varies as a function of $\theta$. In order for the gene expression network to be entrained to the repressilator the global maxima of this protein PSD should be near the repressilator's natural (or peak) frequency of about 1.35 rad/min (see Fig. 3C).

To compute the PSD of the combined network we shall apply Theorem 2.1. For this, we first consider the gene expression network in isolation with $p_2$ in the transcription rate (42) replaced by the constant steady-state mean of $p_2$ (denoted by $\mathbb{E}_\pi(P_2)$). Hence using (40) we can estimate the protein dynamics PSD $S_{X_2}^{iso}(\omega)$ as

$$S_{X_2}^{iso}(\omega) = \frac{2\gamma_p k_p(\theta \mathbb{E}_\pi(P_2) + k_r)}{\gamma_r \gamma_p + k_p k_{fb}}$$

$$\times \left[ \frac{\gamma_r^2 + k_p \gamma_r + \omega^2}{(\gamma_r \gamma_p + k_p k_{fb})^2 + \omega^2(\gamma_r^2 + \gamma_p^2 - 2k_p k_{fb}) + \omega^4} \right].$$

Irrespective of the value of $\theta$, the PSD $S_{X_2}^{iso}(\omega)$ has a global maxima at $\omega_{max} \approx 0.85$ rad/min which is the natural frequency of the gene expression circuit in isolation.

When the repressilator is connected to the gene expression network, we can apply Theorem 2.1 to compute the PSD of the protein output as

$$S_{X_2}(\omega) = S_{X_2}^{iso}(\omega) + \left[ \frac{\theta^2 k_p^2}{(\gamma_r \gamma_p + k_p k_{fb})^2 + \omega^2(\gamma_r^2 + \gamma_p^2 - 2k_p k_{fb}) + \omega^4} \right] S_{cI}(\omega).$$

$$\tag{43}$$

We call this method composite Padé PSD as it estimates the PSD for the full network by combining two PSDs—one obtained with Padé PSD for the nonlinear subnetwork (repressilator) and the other obtained analytically for the linear subnetwork (gene expression). Notably, this method does not require simulations of the combined process, making it easier to obtain PSDs for multiple values of $\theta$ without incurring any simulation burden. In Fig. 5(B) we plot the normalised PSD (area under the PSD curve is normalised to 1) for six values of $\theta$ and we also validate this

composite method with the DFT method for $\theta = 0.4\,\mathrm{min}^{-1}$. One can clearly see that as $\theta$ gets higher, the gene expression network gives up its natural frequency upon stimulation and adopts a frequency which is close to the repressilator frequency. This exemplifies the phenomenon of single-cell entrainment in the stochastic setting.

In order to investigate this entrainment phenomenon further we define an entrainment score as

$$\text{Entrainment Score} = \frac{\int_{\omega_l}^{\omega_r} S_{X_2}(\omega)d\omega}{\int_0^\infty S_{X_2}(\omega)d\omega}, \qquad (44)$$

where $[\omega_l, \omega_r] = [0.9\omega_0, 1.1\omega_0]$ represents an interval of relative length 10% on either side of the repressilator's natural frequency $\omega_0$. In Fig. 5C, we plot a heat-map for the entrainment score as a function of the feedback strength parameter $k_{\mathrm{fb}}$ and the connection strength parameter $\theta$. One can see that the entrainment score increases monotonically with $\theta$ which is to be expected as the first term on the r.h.s. of (43) scales linearly with $\theta$ while the second term scales quadratically. Similarly, by computing the ratio of the two terms we can conclude that entrainment score is also a monotonically increasing function of $k_{\mathrm{fb}}$. However, as the heat-map clearly indicates, the entrainment score is more sensitive to $k_{\mathrm{fb}}$ than $\theta$, thereby suggesting that transcriptional feedback could be a critical mechanism for facilitating entrainment of gene expression networks.

Now suppose that the transcriptional feedback is given by a nonlinear Hill function $F_b(x_2)$. In this case, the gene expression subnetwork becomes nonlinear and Theorem 2.1 cannot be used for PSD estimation. However, we can still employ the Padé PSD method on the combined network using a rational Ansatz of the form (22) with $B(s)$ being the denominator for the Padé approximant estimated by our method in estimating the PSD of the stimulating repressilator network. As shown in Fig. 5D, the PSDs estimated with Padé PSD show good agreement with the DFT-based estimates.

**PSD as a tool for parameter inference**. Consider a self-regulatory gene expression system (see Fig. 6A) modelled as a simple birth-death network where the production rate is given by the repressing Hill function

$$\lambda_H(x) = \frac{K_0}{K_1 + x^H} \qquad (45)$$

of the output copy-number $x$ and the degradation rate is $\gamma$. Fixing all other parameters, our goal is to use the experimental PSD to infer the degree of cooperativity $H$. This experimental PSD is generated via simulations with $H = 1$ and we average the PSDs over 100 single-cell trajectories in order to reduce the variance in the DFT-based PSD estimate. We assume that the experimental single-cell trajectories are proportional to the output copy-number but the constant of proportionality is unknown as is often the case in time-lapse microscopy experiments. We also assume that there is no measurement noise—if the measurement noise appears as an independent process then its PSD simply appears as an additive term in the output PSD, which can be easily removed to recover the output PSD without the measurement noise.

Observe that the unknown constant of proportionality drops out when we compute the normalised PSD (i.e. area under the PSD curve is normalised to 1). Hence we can infer the unknown parameter $H$ by estimating the normalised PSD and comparing it with the experimentally obtained normalised PSD, as was previously demonstrated in[37]. We estimate the normalised PSD with our Padé PSD method and provide a comparison for various values of $H$ in Fig. 6B and it is evident that the experimental traces come from the network with $H = 1$. Note that the clean

estimates for the normalised PSD produced by our Padé PSD method, greatly facilitate the inference of $H$. If the same estimates were obtained with DFT then the estimator noise would obfuscate the dependence of the PSD on $H$ and make the inference task difficult.

**Exploring cell-cycle induced oscillations in gene expression**. In all the examples considered so far, we have ignored that reaction networks reside within cells that are undergoing their own division cycles. Overlooking the cell-cycle is only reasonable when the dynamics of the network being analysed occurs at a timescale which is much faster than the timescale of cell-division. If this assumption does not hold, as is often the case in prokaryotic cells, the cell-cycle process should not be neglected while estimating the frequency spectrum of an output trajectory within a cell-lineage. Tracking trajectories of output fluorescent proteins across a cell-lineage over multiple generations is now increasingly possible due to advanced time-lapse microscopy techniques[70,75] and micro-fluidic platforms such as the mother machine[4]. As these trajectories can be obtained over very long time horizons, a steady-state property like the PSD can be reliably estimated with experimental data, and by comparing it with theoretically estimated PSDs one may gain insights into the underlying network and the role of cell-cycle in inducing oscillations.

Inspired by ref. [37], we consider the cell-cycle evolution as a $N$-stage Markov process with a constant rate $\alpha$ of transitioning from one stage to the next. Hence each transition will occur after a random time-interval which is exponentially distributed with rate $\alpha$. Observe that the expected time to complete one cycle would be $N/\alpha$, implying that the cell-cycle frequency is $f_r = \alpha/N$. At the start of each new cell-cycle, when the cell-cycle process goes from stage $N$ to stage 1, the mother cell undergoes division into two daughter cells and only one of these two cells is tracked and measured, providing us with an output trajectory over a single lineage. The cell division entails a partition of all mother cell molecules into two components—one for each daughter cell. We assume two partitioning mechanisms: symmetric binomial where each mother cell molecule is randomly assigned to each daughter cell with an equal probability, and strict binary where each daughter cell procures exactly half of the mother cell molecules for each network species (see Fig. 7A, C). Observe that partitioning at cell-division forces the displacement in the vector of molecular counts to be state-dependent, i.e. the difference between the state $x$ of the mother cell pre-partition and the state $x'$ of the (tracked) daughter cell post-partition will depend on $x$. Hence, instead of a CTMC with generator (3) we need to model the dynamics with a more general CTMC with generator (4). The explicit form of the generator along with all the computational details on this example can be found in Section S4.2.5 of the Supplement.

Suppose that this dividing cell comprises the gene expression network shown in Fig. 2A which operates at the same timescale as the cell-cycle process. Notice that if we ignore the cell-division cycle, the protein count trajectory does not show any oscillations as seen from the monotonically decreasing PSD plot in Fig. 2A. We now include the cell-cycle and examine how the PSD for the protein counts changes with the cell-cycle length $N$. As we vary $N$ we keep the frequency $f_r$ constant by adjusting $\alpha$. The cell-cycle process can be viewed as an external signal that stimulates the gene expression network by inducing cell-division. Hence we estimate the PSD with our Padé PSD method using a rational Ansatz of the form (22), with $B(s) = |\sigma|^2 - 2\mathrm{Real}(\sigma)s + s^2$ where $\sigma = -\alpha(1 - \exp(2\pi i/N))$ is the eigenvalue of the cell-cycle evolution generator with the least magnitude of the real part. The estimated PSDs show good agreement with the PSDs estimated via DFT, for both types of partitioning mechanisms

(see Fig. 7B, D) and one can see that the type of partitioning mechanism has little effect on the PSD. Moreover as $N$ increases, the relative noise in the cell-cycle process goes down, causing an increase in the off-zero peak of the PSD at the cell-cycle frequency of roughly 1.57 rad/min. This observation is consistent with the results reported in ref. [37] for a single-species bursty gene expression network but with a much richer cell-division model than what we consider. The analytical computations presented in ref. [37] are quite elegant and the authors employ generating function techniques to obtain closed-form expressions for the PSD under the assumption of binomial partitioning. However, this analytical approach may become infeasible when other partitioning mechanisms are considered (e.g. strict binary) or when the output trajectories come from a high-dimensional nonlinear network. Our numerical Padé PSD method should still perform reliably in these cases as long as one can feasibly simulate the stochastic trajectories of the process.

## Discussion

Recent advances in microscopic imaging and fluorescent reporter technologies have enabled high-resolution monitoring of processes within living cells[5]. As the accessibility of this time-course data rapidly increases, there is an urgent need to design theoretical and computational approaches that make use of the full scope of such data, in order to understand intracellular processes and design effective synthetic circuits. An important feature of time-course measurements, which is lacking in the data generated by the more common experimental technique of Flow-Cytometry, is that they capture temporal correlations at the single-cell level which are rich in information about the underlying dynamical model. Frequency-domain analysis provides a viable approach to extract this information, if we have an efficient framework to connect network models to the frequency spectrum or the power spectral density (PSD) of the single-cell trajectories measured with time-lapse microscopy[18,20]. The dynamics within cells is invariably stochastic, owing to the presence of many low abundance biomolecular species, and it is commonly described as a continuous-time Markov chain (CTMC). In this context, the aim of this paper is to develop a computational method for reliably estimating the PSD for single-cell trajectories from CTMC models. Existing approaches for PSD estimation for stochastic network models, are either applicable to a particular class of networks[17,26], or they are based on dynamical approximations that are known to be inaccurate over large time-intervals and in situations where low abundance species are present[19,20]. The method we develop in this paper, called Padé PSD, especially pertains to the low abundance regime. It applies generically to any stable network and it yields an accurate PSD expression using a small number of CTMC trajectory simulations. Moreover, for networks with affine propensity functions, we provide a PSD decomposition result that expresses the output PSD in terms of its constituent parts.

The tools we develop in this paper are of significance to both systems and synthetic biology. We demonstrate that in the presence of intrinsic noise, PSD estimation can successfully differentiate between adapting Incoherent Feedforward (IFF) and Negative Feedback (NFB) topologies[34], and it can facilitate performance optimisation of synthetic oscillators[35] as well as synthetic in vivo controllers[36]. Moreover, it can also aid the study of stochastic entrainment at the single-cell level. This is of particular relevance for applications such as designing pulsatile dynamics of transcription factors, which is known to enable graded multi-gene regulation[76]. We present a simple nonlinear network to illustrate that PSDs enable parameter inference from experimental single-cell trajectory data without requiring the explicit knowledge of the

constant of proportionality that links the output species copy-number to the observed signal. Lastly, we consider an example with cell-division cycles and show that our Padé PSD method provides accurate PSD estimates for stochastic trajectories from a single lineage, thereby assisting in precise quantification of the oscillations induced by the cell-cycle process.

The main contribution of this paper is to show how the theory of Padé approximations can be effectively applied to the PSD estimation problem for reaction networks with stochastic CTMC dynamics. In Padé PSD a low dimensional approximation of the PSD is computed based on estimates of Padé derivatives that are expressible as certain stationary expectations for which efficient Monte Carlo estimators were developed. As our method requires simulations of stochastic trajectories it naturally inherits the associated drawbacks—these simulations can be computationally expensive, especially if the network possesses multiple reaction time-scales. Fortunately, the problem of reliably estimating expectations under the CTMC model has received a lot of attention in recent years[77], and various methods designed for this problem, like $\tau$-leaping[78] and/or multilevel schemes[79], can be easily integrated with Padé PSD, in order to speed up the estimation process and also to reduce the variance of the Monte Carlo estimators. Moreover, model reductions[80,81] and simulation tools[82,83] for multiscale networks can be readily applied to simplify the estimation of Padé derivatives. Such extensions would greatly expand the scope of applicability of our method and pave the way for frequency-based analysis and design of stochastic biomolecular reaction networks.

## Methods

We now discuss the computational implementation of our Padé PSD method. The detailed algorithms for this method are provided in Section S3 of the Supplement and its full `Python` implementation is available on GitHub: https://github.com/ankitgupta83/PadePSD_python.git[84].

The inputs to our method are as follows:

- A positive integer $p$ which specifies the order of the rational Padé approximant $G_p(s)$ given by (15).
- A vector of distinct points $\mathbf{s} = (s_1, \ldots, s_L)$ on the extended positive real-line $(0, \infty]$ along with a vector of positive integers $\boldsymbol{\rho} = (\rho_1, \ldots, \rho_L)$. The Padé approximant is constructed by matching between $G(s)$ and $G_p(s)$ the first $\rho_\ell$ terms in the power series expansion around $s = s_\ell$ for each $\ell = 1, \ldots, L$. Without losing any generality we may assume that $s_1, \ldots, s_{L-1}$ are all finite and $s_L = \infty$.
- A vector of distinct positive real test values $\bar{\mathbf{s}} = (\bar{s}_1, \ldots, \bar{s}_R)$ for validating the Padé approximant.

Given these inputs, the main computational tasks that Padé PSD performs are:

1. *Estimate the required Padé derivatives*: Quantities $D_m^{(s_\ell)}$ are estimated for each $m = 0, 1, \ldots, (\rho_\ell - 1)$ and each $\ell = 1, \ldots, L$.
2. *Obtain direct estimates for validation*: Quantities $(G(\bar{s}_1), \ldots, G(\bar{s}_R))$ are directly estimated.

Upon completing these tasks, the linear system (21) for the $2p$ coefficients for the Padé approximant $G_p(s)$ is constructed and solved. This provides us with $G_p(s)$ which is then validated with the direct estimates $(G(\bar{s}_1), \ldots, G(\bar{s}_R))$, and if the validation is successful, the PSD $S_{X_n}(\omega)$ is obtained by applying formula (9) with $G(z) = G_p(z)$.

All the required quantities are simultaneously estimated with $Q$ trajectories of the augmented CTMC $(\mathcal{X}(t))_{t \geq 0}$ with

$$\mathcal{X}(t) = (X(t), Y(t), Z(t))$$

where

- $X(t) = (X_1(t), \ldots, X_d(t))$ is the vector of species copy-numbers.
- $Y(t) = (Y_1(t), \ldots, Y_{\vartheta_1}(t), Y_{\vartheta_1+1}(t), \ldots, Y_{\vartheta_2}(t), \ldots, Y_{\vartheta_{L-1}+1}(t), \ldots, Y_{\vartheta_{L-1}}(t), \ldots,$ is the vector of additional state-components used for estimating the Padé derivatives $D_m^{(s_\ell)}$ for each $m = 0, 1, \ldots, (\rho_\ell - 1)$ and each $\ell = 1, \ldots, (L-1)$. Here $\vartheta_\ell = \sum_{j=1}^{\ell} \rho_j$ with $\vartheta_0 = 0$. Note that the estimation of the Padé derivatives at $s_L = \infty$ does not require these additional state components.
- $Z(t) = (Z_1(t), \ldots, Z_R(t))$ is the vector of additional state-components used for estimating $(G(\bar{s}_1), \ldots, G(\bar{s}_R))$.

The augmented process has

$$\underbrace{K}_{\text{original network reactions}} + \underbrace{L-1}_{\mathcal{R}_{s_1},\ldots,\mathcal{R}_{s_{L-1}}} + \underbrace{R}_{\mathcal{R}_{\bar{s}_1},\ldots,\mathcal{R}_{\bar{s}_R}}$$

reactions. Note that each reaction $\mathcal{R}_s$ has the constant propensity of $s$. Our Padé PSD method simulates such a reaction network over the time-interval $[0, T_f]$, by extending the classical Gillespie's Stochastic Simulation Algorithm[39], and then estimates the Padé derivatives and the direct estimates $(G(\bar{s}_1), \ldots, G(\bar{s}_R))$. Under this extension, when the firing reaction is $k = 1, \ldots, K$, then the state $(x, y, z)$ moves to $(x + \zeta_k, y, z)$ as in the original CTMC. However, when the firing reaction is $\mathcal{R}_{s_\ell}$ for some $\ell = 1, \ldots, (L-1)$ then the state $(x, y, z)$ moves to $(x, y', z)$ where

$$y'_j = \begin{cases} x_n & \text{if } j = \vartheta_{\ell-1} + 1 \\ y_{j-1} & \text{if } j = \vartheta_{\ell-1} + 2, \ldots, \vartheta_\ell \\ y_j & \text{otherwise.} \end{cases} \quad (46)$$

Similarly, if the firing reaction is $\mathcal{R}_{\bar{s}_r}$ for some $r = 1, \ldots, R$ then the state $(x, y, z)$ moves to $(x, y, z')$ where

$$z'_r = x_n \text{ and } z'_j = z_j \text{ for all } j \neq r. \quad (47)$$

Estimation of the Padé derivatives at $\infty$ (i.e. $D_m^{(\infty)}$ for $m = 0, \ldots, (\rho_L - 1)$) requires several evaluations of functions of the form $\mathbb{A}^m f(x)$. This can be done recursively but it is computationally very intensive. In order to minimise these evaluations we exploit the fact that ergodic Markov chains visit the same set of states again and again. Therefore if we can intelligently store the values $\mathbb{A}^m f(x)$ generated by this function, and quickly retrieve them as needed, then it provides a way to leverage the vast memory resources in modern computers in order to gain computational efficiency. Fortunately, `Python` provides an ideal data structure, called a `dictionary`, for this purpose and we use it in our computational implementation to boost the efficiency of Padé PSD.

**Reporting summary**. Further information on research design is available in the Nature Research Reporting Summary linked to this article.

## Data availability

Custom code was written in Python for data generation. This code is publicly available at the indicated GitHub repository[84].

## Code availability

The Python code for data generation and analysis can be downloaded from the GitHub repository: https://github.com/ankitgupta83/PadePSD_python.git[84].

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

## Acknowledgements

This project has received funding from the European Research Council (ERC) under the European Union's Horizon 2020 research and innovation programme grant agreement no. 743269 (CyberGenetics project), and from the Swiss National Science Foundation under grant number 182653.

## Author contributions

A.G. and M.K. conceived the project; A.G. performed the theoretical and computational analysis with inputs from M.K.; A.G. and M.K. wrote the paper; M.K. secured the funding.

## Competing interests

The authors declare no competing interests.
