## [Peer Review File · Nature Communications]

Reviewers' Comments:

Reviewer #1:

Remarks to the Author:

Gupta and Khammash present an interesting paper on the utility of frequency spectra in systems biology. They argue that the application of analytical techniques in the frequency domain has been much more limited in biology than in other disciplines. They are in particular concerned with the use of these methods to obtain insight into the frequency content of noisy single-cell trajectories. They develop a method which approximates the power spectrum via Pade approximations from a handful of trajectory simulations. They illustrate the technique on various cases.

I generally have a favourable opinion of this manuscript. Indeed I concur with the authors that there are few papers exploring the frequency content of single-cell trajectories and that a lot of information is in them potentially waiting to be uncovered. The Pade approximation they devise also has a robust theoretical foundation and does well in the various examples they show. What however I am not so convinced about is its utility to actual data because of inherent assumptions made in the modelling framework; as well there is in the past year published work which goes in the same direction albeit using different methods and which has been already shown to be very useful on actual data. My more detailed comments are as follows:

1. As a motivation for their study, it is mentioned in several places through the manuscript, that temporal correlations are a feature of single-cell trajectories and that many current methods do not allow us to extract this information because they are designed to give information about a population snapshot in time, e.g. using flow cytometry. I agree with this statement. However the issue is that from what is presented, it is not clear that their method can capture temporal correlations in actual single-cell trajectories. These trajectories, e.g. lineage data obtained from mother machines (e.g. Tanouchi et al. "A noisy linear map underlies oscillations in cell size and gene expression in bacteria." *Nature* 523.7560 (2015): 357-360), are typically cyclic in nature due to the quasi-periodic nature of cell division. The noise from this process and from cell-cycle processes, e.g. replication, is known to be crucial to describing such trajectories since one is following a single cell across its cell cycle. Now to capture such details, the chemical master equation has to be augmented with (at least) terms that can describe binomial partitioning due to cell division; this term is not considered in this paper. All the examples implicitly assume that division can be effectively modelled via a first-order decay reaction but this has been shown to not be a good approximation in the vast majority of cases. See for e.g. Beentjes et al. "Exact solution of stochastic gene expression models with bursting, cell cycle and replication dynamics." *Physical Review E* 101.3 (2020): 032403. Hence it is crucial and within the scope of the manuscript to show that the Pade approximation works when the master equation is extended to describe at least cell division explicitly because otherwise their results are irrelevant to single-cell trajectories from lineage data. The authors need to introduce a relevant example and compare the approximation vs simulations. For e.g. they can consider the extension of the reaction scheme in Fig. 2A to allow for binomial partitioning of mRNA and proteins, assuming a cell-cycle duration of fixed length (for simplicity).

2. I also wish to bring to the attention of the authors that a recent paper has derived exact expressions for the power spectra of single-cell trajectories in a detailed model of gene expression that takes into account a large degree of biological realism, e.g. replication, division, dosage compensation, cell size dynamics, bursty expression and cell-cycle duration variability.

Jia and Grima. "Frequency domain analysis of fluctuations of mRNA and protein copy numbers within a cell lineage: theory and experimental validation." *Physical Review X* 11.2 (2021): 021032.

This work is highly relevant to what is presented in this paper because its aim is also to extract information from the power spectrum of single cell trajectories and the authors show its application to real data as well. Also it is shown that parameters can be estimated without the explicit knowledge of the proportionality constant relating the fluorescent intensity to the copy-number, a claim that is also mentioned in the present manuscript. Hence there is a fair degree of intersection between the present paper and the above one, not technically, but in terms of aims. It is important that in the Conclusions, the authors contrast their Pade approximation method via the

above, clearly discussing but the success and the limitations of the two approaches vis-a-vis each other.

3. I noticed that none of the examples considered show any noise-induced oscillations, i.e. where the deterministic has no sustained oscillations but the stochastic model shows a power spectrum peaked at a non-zero value of the frequency. These examples have been well studied using various methods (as the authors also acknowledge) and are common in nature. As well, because the noise here is what constructs the oscillations, these features of the power spectrum are typically more difficult to predict than in cases where there are deterministic oscillations. Hence I would like the authors to investigate one case where there are noise-induced oscillations and to show that their Padé approximation works well here too.

Reviewer #2:

Remarks to the Author:

In the paper 'Frequency Spectra and the Color of Cellular Noise' the authors Ankit Gupta and Mustafa Khammash investigate the Power Spectral Density of Chemical Reaction Networks.

For transparency, I want to remark that I reviewed this manuscript already in an earlier version. Some of my review remarks were incorporated and the manuscript improved. In particular, a new Padé approximation is presented. Some of my general concerns about the paper however remain and I am therefore restating or adapting them below.

In section 2, the authors introduce their main tool, the resolvent operator. After a short discussion of the consequences for linear chemical reaction networks (section 3), they proceed with their main method, the Padé PSD, in section 4. Case studies are given in section 5.

The authors verify that the Padé approximation is close to the exact PSD for the linear examples (5.1). While I consider the Padé PSD innovative, my major criticism is directed towards the applicability of the theory to the currently available data of biological systems. In the introduction the theory on CRNs is portrayed as lagging behind experimental capabilities. I do not agree and think the opposite is true. In particular in the context of noise spectra, I want to raise doubts whether the short time series measured in real biological systems can inform accurate noise spectra. The 2005 (!) reference [2] used to support the claim that high-resolution techniques are now available seems inappropriate; fluorescent proteins are rather elementary in today's research. There has been indeed ground-breaking progress in microscopy such as for instance MINIFLUX, but those techniques rather fall into the class of super-resolution techniques. The strength of the paper clearly lies in abstract statements about the qualitative behavior of CRN models, e.g. network topology distinction via existence of a local maximal in the PSD (5.2). This said, the portrayal as being directed towards application in the experimental context seems misleading. Additionally, the majority of applications in the paper relies on linear CRNs or linearized synthesis rate which may not do justice to the biological reality. In general, a claim that a new computational tool is so essential for experimental work logically requires a demonstration of the method on experimental data. That essential part is missing.

Hence, it is a technically sound contribution that may fit well to a computationally-oriented journal such as JCP but in my opinion it does not reach the significance and broad interest level targeted by Nat Comm.

More technical remarks:

The authors emphasize the exactness of their results as a novelty in contrast to approximations via the Chemical Langevin equation (CLE) at several places (p. 3, lines 5-7, 11-13, 4th paragraph last sentence). However, for linear CRNs which constitute a major part of their applications, second order statistics agree with those obtained by the corresponding CLE [14].

The theorem 3.1. is formulated comprehensibly and appeals by its interpretability. Its usefulness for application in the study of cell-to-cell heterogeneity is apparent. But its proof in S.2.3 seems

overly complex considering that the rather intuitive statement appeals as a form of covariance decomposition. It seems not instructive in that it provides a better understanding of the Pade PSD. Overall, the transfer from results on CLE to CTMC should find much higher mentioning in the paper (e.g. p.16, lines 4-5 are not surprising with this background).

Comments on Examples in Section 5:

Eq (35), (36) were already covered in [14], Raj 2006 "Stochastic mRNA Synthesis in Mammalian Cells". This is because noise power spectra find frequent application in the literature for determining the variance (e.g. Raj2006) via the computation of the total stationary variance as the integral over the noise power spectrum as described in Box 1.

There is a typo in (37): It should read $2k_{on}k_{off}/(k_{on} + k_{off})((k_{on} + k_{off})^2 + w^2)$

Example (37), (38) is covered in [14] with minor modifications. Should there not be another degradation rate for the unspliced?

The application of PSD as a way of distinguishing between network architectures in 5.2. is convincing.

In Fig. 2C degradation of C is missing in upper panel. Linearizing the function in $x_C = 0$, assumes that x_C assumes small values. However, as seen in the figure, x_C assumes large values.

Both IFF and NFB are portrayed in the high copy number regime which does not support well the authors' distinct focus on small copy numbers and their rejection of the CLE, which as mentioned by the authors would not change the results anyways.

(NFB) Fig. 2D upper right the red inhibition should point toward the arrow from I to C.

An argument is missing why the qualitative behaviour (existence of oscillation), that was obtained from linearized F_b , should persist if the positive part of F_b is used, as was done. Since the mean value \bar{x}_o is close to the zero of F_b , this becomes rather important. The CLE analysis can handle negative rates but changing to a non-linear F_b (by taking the positive part) may alter the behaviour.

Minor objections in the first part:

- The statement that an SDE approximation is only good for finite time (p.3, lines 8-10), not at stationarity, needs further explanation.
- (3) is the adjoint operator
- Eq (35): Why is there a reference to the Cauchy distribution? $S_X(w)$ is not a density of a measure.

As far as the Pade PSD part is concerned, I am neither familiar with the Padé approximation nor with the current state of research on PSD approximations. From that perspective, the approach is new to me and seems innovative. 5.3 and 5.4. discuss interesting applications in synthetic biology, where the method can help to identify parameter regimes that are predicted to yield desired outcomes (stabilized oscillations or oscillation reduction) when used in experiments.

The authors do not clearly discuss drawbacks of the method. Vaguely, high computational costs are mentioned in the discussion leaving room for improvement.

Response Letter

Frequency Spectra and the Color of Cellular Noise

Ankit Gupta^{*1} and Mustafa Khammash^{†1}

¹Department of Biosystems Science and Engineering, ETH Zürich, Mattenstrasse 26, 4058 Basel, Switzerland.

March 24, 2022

We thank the reviewers of *Nature Communications* for going through our paper and making many important remarks. We have revised our manuscript accordingly and below we describe all the changes in detail. The parts of the paper that have been added/modified appear in **blue** to allow the reviewers to locate these parts easily. We first mention all the major changes that have been made to the manuscript and then we provide a detailed response to each of the reviewers' comments.

Major Changes

- A new and improved method:** The main contribution of our paper is the development of a computational method called *Padé PSD* that estimates the PSD for stochastic single-cell trajectories as a rational function of a specific form. In the previous version, this rational function was identified with *two-point* Padé approximation that matches a certain number of terms in the power series expansions at two points (which were chosen to be ∞ and some positive number s_0). In the revised version of the manuscript we have extended this method to a more general *multipoint* Padé approximation scheme that allows power series matching at several points s_1, \dots, s_L on the extended positive real-line $(0, \infty]$. This makes our PSD estimation more robust, especially is dealing with networks with complex PSDs.
- Better estimators for the Padé derivatives:** The power series coefficients that are matched by *Padé PSD* are called Padé derivatives and they need to be estimated via simulations. We have modified the design of these estimators (for finite values of s_ℓ) in order to improve their statistical accuracy and this improvement is reflected in our numerical results.

*ankit.gupta@bsse.ethz.ch

†mustafa.khammash@bsse.ethz.ch

3. **Allowing more general CTMC dynamics:** Previously we were working with the standard CTMC-based stochastic model of a reaction network, where each reaction k causes a fixed displacement ζ_k (the stoichiometric vector) in the state-vector $x = (x_1, \dots, x_d)$ of molecular counts. In this model the displacement cannot be random and it cannot depend on the state x . Hence, as pointed by Reviewer 1, it cannot handle cell-wide transitions like cell-division that can make the state x jump by a random amount (when one tracks cells along a single-lineage) that will certainly depend on the state x . To allow such transitions, in the revised manuscript we have updated *Padé PSD* to work with a more general CTMC model that incorporates random state-dependent transitions. This enables us to add the example that we mention next.
4. **A new example showing cell-cycle induced oscillations:** As suggested by Reviewer 1, we include cell-cycle evolution and cell-division in one of our examples (the gene-expression network). With our updated *Padé PSD* method, we analyse how cell-cycle dynamics induces oscillations in an otherwise non-oscillating network and similar to the results reported in [1] we demonstrate that the relative power of the oscillatory component at the cell-cycle frequency rises as the variability of the cell-cycle evolution goes down.
5. **Other additions to the previous examples:** As mentioned by Reviewer 1, generally it is difficult to estimate PSD for networks with *noise-induced* oscillations, i.e. where the deterministic trajectories converge to a fixed point but the stochastic trajectories exhibit sustained oscillations. For the example with the antithetic controller, we have added new results to show that this network does indeed exhibit noise-induced oscillations and our method is quite accurate in estimating the PSD for the stochastic model. In the example with the repressilator stimulating a gene-expression network with feedback, we previously only considered linearised feedback and studied the entrainment phenomenon. In the revised manuscript we have also added a couple of plots showing the accuracy of our method in estimating the PSDs when the feedback in the gene-expression network is modelled by a nonlinear Hill function.

Response to Reviewer 1

1. *Gupta and Khammash present an interesting paper on the utility of frequency spectra in systems biology. They argue that the application of analytical techniques in the frequency domain has been much more limited in biology than in other disciplines. They are in particular concerned with the use of these methods to obtain insight into the frequency content of noisy single-cell trajectories. They develop a method which approximates the power spectrum via Pade approximations from a handful of trajectory simulations. They illustrate the technique on various cases.*

I generally have a favourable opinion of this manuscript. Indeed I concur with the authors that there are few papers exploring the frequency content of single-cell trajectories and that a lot of information is in them potentially waiting to be uncovered. The Pade approximation they devise also has a robust theoretical foundation and does well in the various examples they show. What however I am not so convinced about is its utility to

actual data because of inherent assumptions made in the modelling framework; as well there is in the past year published work which goes in the same direction albeit using different methods and which has been already shown to be very useful on actual data. My more detailed comments are as follows:

Answer: We thank the reviewer for a favourable opinion of our manuscript. As will be evident from our responses to the reviewer’s comments, we have made many changes to address the concerns of the reviewer.

- 2. As a motivation for their study, it is mentioned in several places through the manuscript, that temporal correlations are a feature of single-cell trajectories and that many current methods do not allow us to extract this information because they are designed to give information about a population snapshot in time, e.g. using flow cytometry. I agree with this statement. However the issue is that from what is presented, it is not clear that their method can capture temporal correlations in actual single-cell trajectories. These trajectories, e.g. lineage data obtained from mother machines (e.g. Tanouchi et al. "A noisy linear map underlies oscillations in cell size and gene expression in bacteria." Nature 523.7560 (2015): 357-360), are typically cyclic in nature due to the quasi-periodic nature of cell division. The noise from this process and from cell-cycle processes, e.g. replication, is known to be crucial to describing such trajectories since one is following a single cell across its cell cycle. Now to capture such details, the chemical master equation has to be augmented with (at least) terms that can describe binomial partitioning due to cell division; this term is not considered in this paper. All the examples implicitly assume that division can be effectively modelled via a first-order decay reaction but this has been shown to not be a good approximation in the vast majority of cases. See for e.g. Beentjes et al. "Exact solution of stochastic gene expression models with bursting, cell cycle and replication dynamics." Physical Review E 101.3 (2020): 032403. Hence it is crucial and within the scope of the manuscript to show that the Pade approximation works when the master equation is extended to describe at least cell division explicitly because otherwise their results are irrelevant to single-cell trajectories from lineage data. The authors need to introduce a relevant example and compare the approximation vs simulations. For e.g. they can consider the extension of the reaction scheme in Fig. 2A to allow for binomial partitioning of mRNA and proteins, assuming a cell-cycle duration of fixed length (for simplicity).*

Answer: We have followed the reviewer’s advice and extended our CTMC framework to include jump terms that correspond to partitioning at cell-division. The partitioning scheme can be freely chosen and it need not be binomial. We have included the example suggested by the reviewer that extends the gene-expression network to allow for cell-cycle evolution and cell-division. We show that our method is able to accurately estimate the PSD of the single-cell trajectories from a single lineage.

- 3. I also wish to bring to the attention of the authors that a recent paper has derived exact expressions for the power spectra of single-cell trajectories in a detailed model of gene expression that takes into account a large degree of biological realism, e.g. replication, division, dosage compensation, cell size dynamics, bursty expression and cell-cycle duration variability.*

Jia and Grima. "Frequency domain analysis of fluctuations of mRNA and protein copy numbers within a cell lineage: theory and experimental validation." *Physical Review X* 11.2 (2021): 021032.

This work is highly relevant to what is presented in this paper because its aim is also to extract information from the power spectrum of single cell trajectories and the authors show its application to real data as well. Also it is shown that parameters can be estimated without the explicit knowledge of the proportionality constant relating the fluorescent intensity to the copy-number, a claim that is also mentioned in the present manuscript. Hence there is a fair degree of intersection between the present paper and the above one, not technically, but in terms of aims. It is important that in the Conclusions, the authors contrast their Padé approximation method via the above, clearly discussing but the success and the limitations of the two approaches vis-a-vis each other.

Answer: We thank the reviewer for bringing this excellent and highly relevant paper to our attention. We cite this paper at multiple places in our revised manuscript and also borrow the model for cell-cycle evolution from it. The analyses in [1] makes a very elegant use of generating functions and they provide expressions for the PSD for a single-species gene-expression model with many biologically pertinent features like replication, dosage compensation, bursty expression etc. However this analytical approach may not work if non-binomial partitioning is considered, or if the gene-expression network is more complex with multiple species. On the other hand, our numerical procedure *Padé PSD* can easily handle such generalities, but of course it does not provide analytical expressions like the results in [1]. We comment on this while discussing the example of a two-species gene-expression model with cell-cycle.

4. *I noticed that none of the examples considered show any noise-induced oscillations, i.e. where the deterministic has no sustained oscillations but the stochastic model shows a power spectrum peaked at a non-zero value of the frequency. These examples have been well studied using various methods (as the authors also acknowledge) and are common in nature. As well, because the noise here is what constructs the oscillations, these features of the power spectrum are typically mode difficult to predict that in cases where there are deterministic oscillations. Hence I would like the authors to investigate one case where there are noise-induced oscillations and to show that their Padé approximation works well here too.*

Answer: This is a good point and we have added new results in the example with a gene-expression network controlled by the antithetic controller. We show that this network does indeed exhibit noise-induced oscillations and our method is quite accurate in estimating its PSD.

Response to Reviewer 2

1. In the paper ‘Frequency Spectra and the Color of Cellular Noise’ the authors Ankit Gupta and Mustafa Khammash investigate the Power Spectral Density of Chemical Reaction Networks.

For transparency, I want to remark that I reviewed this manuscript already in an earlier version. Some of my review remarks were incorporated and the manuscript improved. In particular, a new Pade approximation is presented. Some of my general concerns of about the paper however remain and I am therefore restating or adapting them below.

In section 2, the authors introduce their main tool, the resolvent operator. After a short discussion of the consequences for linear chemical reaction networks (section 3), they proceed with their main method, the Pade PSD, in section 4. Case studies are given in section 5.

The authors verify that the Padé approximation is close to the exact PSD for the linear examples (5.1). While I consider the Pade PSD innovative, my major criticism is directed towards the applicability of the theory to the currently available data of biological systems. In the introduction the theory on CRNs is portrayed as lagging behind experimental capabilities. I do not agree and think the opposite is true. In particular in the context of noise spectra, I want to raise doubts whether the short time series measured in real biological systems can inform accurate noise spectra. The 2005 (!) reference [2] used to support the claim that high-resolution techniques are now available seems inappropriate; fluorescent proteins are rather elementary in today’s research. There has been indeed ground-breaking progress in microscopy such as for instance MINFLUX, but those techniques rather fall into the class of super-resolution techniques. The strength of the paper clearly lies in abstract statements about the qualitative behavior of CRN models, e.g. network topology distinction via existence of a local maximal in the PSD (5.2). This said, the portrayal as being directed towards application in the experimental context seems misleading. Additionally, the majority of applications in the paper relies on linear CRNs or linearized synthesis rate which may not do justice to the biological reality. In general, a claim that a new computational tool is so essential for experimental work logically requires a demonstration of the method on experimental data. That essential part is missing.

Hence, it is a technically sound contribution that may fit well to a computationally-oriented journal such as JCP but in my opinion it does not reach the significance and broad interest level targeted by Nat Comm.

Answer: We thank the reviewer for going through our manuscript again. We hope that the reviewer would find the updated manuscript more suitable for publication in Nature Communications.

The reviewer raises a very valid concern of whether our method can help in connecting theoretical models with experimental data if the trajectories cannot be measured over long time periods. Firstly, we would like to point out that even short time-trajectories might be enough if the underlying network operates at a much faster time-scale. Secondly, with modern-day microscopy, imaging and microfluidic techniques, it is possible

to measure the trajectory from a lineage of a dividing cell-population over multiple generations (see [2] for a recent review). In fact even as far back as 2013, researchers have been able to obtain trajectories from bacterial cells for more than 7 days (> 350 generations) [3], and the technologies have improved significantly since 2013. As our newly added example with cell-division cycles shows (see Section 5.7), our method can be used to estimate PSDs for such single-cell lineage data.

When we highlight the need for developing theoretical and computational frequency domain tools, it is motivated by the fact that there are few computational methods for reliable PSD estimation for general nonlinear stochastic reaction networks, and this hinders adoption of frequency-based analysis in systems and synthetic biology and connection with power spectrums of experimental single-cell trajectories. This assertion is supported by Reviewer 1 and our goal in this paper is to address this issue and come up with a tractable tool for PSD estimation which does not suffer from the same issues as the standard DFT-based method for PSD estimation (i.e. high estimator noise and inconsistency, aliasing effects etc.). It must be noted that after an efficient numerical approach called Finite State Projection (FSP) [4] was developed for solving the CME, it was rapidly adopted by the research community and many biologically significant studies used it in conjunction with experimental population snapshot data from techniques such as Flow-Cytometry and single-molecule fluorescent in situ hybridization (smFISH) (e.g. see [5, 6, 7]). In the same vein, we hope that our method for PSD estimation would similarly provide a way for researchers to meaningfully utilise time-lapse imaging data which is rich in information about the underlying network.

With regards to how our work can be connected to experimental data, an example of that can be found in a recent paper from our research group [8] where trajectories of nascent RNA counts in engineered yeast cells were measured (for > 300 minutes), and the frequency spectrum was heavily used in assessing the performance of various synthetic controllers. The analysis reported in Section 5.4 of this paper showed that adding an extra proportional feedback reduces the oscillations created by the antithetic controller, and motivated by this finding experiments were carried out with the *Cyber-loop* platform and this finding was verified with single-cell trajectory data in [8] (see Figure 1 in this document).

The comment on linear CRNs is addressed in our response to the next point.

In light of the reviewer’s main criticism we have revised the introduction of the paper to provide more relevant references highlighting that long-term single-cell trajectories can indeed be experimentally obtained across a single-lineage.

2. *More technical remarks:*

The authors emphasize the exactness of their results as a novelty in contrast to approximations via the Chemical Langevin equation (CLE) at several places (p. 3, lines 5-7, 11-13, 4th paragraph last sentence). However, for linear CRNs which constitute a major part of their applications, second order statistics agree with those obtained by the corresponding CLE [14].

Answer: We agree that for linear networks CLE provides the exact PSD and we mention this explicitly in the revised manuscript (see pages 3 and 7). Here by CLE the

Figure 1: This figure is taken from Figure 5 in [8] and it shows reduction of oscillations caused by the antithetic controller upon the addition of extra proportional negative feedback. $|P(\omega)|$ is a proxy for the PSD, AIC is the standard antithetic integral controller and APIC is this controller with the extra proportional feedback.

authors in [9] refer to what we call the Linear Noise Approximation (LNA) as the noise terms are constructed using the average propensities rather than the instantaneous propensities (see the Appendix of [9]). It is known that for nonlinear networks the LNA provides a very inaccurate PSD (see [10]), and this emphasises the need for a method like Padé PSD. This inaccuracy of LNA can also be adjudged by our newly-added example of the antithetic controller that operates in a regime where the deterministic model exhibits convergence to a fixed point but the stochastic model exhibits sustained oscillations, thereby showing that the LNA cannot be accurate (see our response to point 7).

We would like to emphasise that linear CRNs do not constitute a major part of the applications that we present. They are mainly used for either validating our Padé PSD method or for demonstrating our PSD decomposition result (which is novel). In the study where we show that PSDs can distinguish between architectures for adapting networks, the linearisation was an analytical tool that allowed us to apply our decomposition result and our conclusions do not depend on the parameters of the linearised network or the point of linearisation. Most of the applications we provide, rely on applying our Padé PSD method on nonlinear networks like the repressilator, the antithetic integral feedback controller, gene-expression with nonlinear autoinhibition etc. In the revised manuscript we have added new results with nonlinear networks. These include an example showing cell-cycle induced oscillations and a nonlinear gene-expression network driven by the repressilator.

3. *The theorem 3.1. is formulated comprehensibly and appeals by its interpretability. Its usefulness for application in the study of cell-to-cell heterogeneity is apparent. But its proof in S.2.3 seems overly complex considering that the rather intuitive statement*

appeals as a form of covariance decomposition. It seems not instructive in that it provides a better understanding of the Padé PSD. Overall, the transfer from results on CLE to CTMC should find much higher mentioning in the paper (e.g. p.16, lines 4-5 are not surprising with this background).

Answer: We thank the reviewer for acknowledging the usefulness of the PSD decomposition result. Indeed, its proof in the Supplement is complicated because we are considering a general linear network being stimulated by a general signal (which may not come from a linear network). For the proof to work, several steps are required and this makes the proof span around four pages. To put this in context, an even longer proof of this result is given in a recent paper [11] for the special case of the linear network being a simple single-species birth-death model. To make the proof more accessible we have provided a summary of all the steps in the revised version of the supplement to the manuscript. Note that this PSD decomposition result for linear networks is not connected to our Padé PSD method and it is not included to provide a better understanding of the Padé PSD method.

For the remark on transfer of results from CLE to CTMC, please see our response to the previous point.

4. *Comments on Examples in Section 5: Eq (35), (36) were already covered in [14], Raj 2006 “Stochastic mRNA Synthesis in Mammalian Cells”. This is because noise power spectra find frequent application in the literature for determining the variance (e.g. Raj2006) via the computation of the total stationary variance as the integral over the noise power spectrum as described in Box 1. There is a typo in (37): It should read $2k_{on} * k_{off} / (k_{on} + k_{off}) ((k_{on} + k_{off})^2 + w^2)$ Example (37), (38) is covered in [14] with minor modifications. Should there not be another degradation rate for the unspliced? The application of PSD as a way of distinguishing between network architectures in 5.2. is convincing.*

Answer: Thanks for pointing out the typo. We have corrected it and also added the degradation rate for the unspliced mRNA. We agree that expressions (35) and (36) were present in earlier works. Our aim in presenting simple linear networks is not to provide expressions for their PSD (which are already known) but to illustrate our novel PSD decomposition result and validate our Padé PSD method. We thank the reviewer for finding the application of PSDs in distinguishing architectures convincing. This application relies on the PSD decomposition result which allows us to systematically modularise the PSD computation.

5. *In Fig. 2C degradation of C is missing in upper panel. Linearizing the function in $x_C = 0$, assumes that x_C assumes small values. However, as seen in the figure, x_C assumes large values.*

Answer: We are not linearising around $x_c = 0$ but around an arbitrary point $x_c = x_0$. Note that

$$F_f(x_c) = \beta_1 - \beta_{ff}(x_c - x_0) = \beta_1 + \beta_{ff}x_0 - \beta_{ff}x_c = \beta_0 - \beta_{ff}x_c,$$

where $\beta_0 = \beta_1 + \beta_{ff}x_0$.

6. Both IFF and NFB are portrayed in the high copy number regime which does not support well the authors' distinct focus on small copy numbers and their rejection of the CLE, which as mentioned by the authors would not change the results anyways. (NFB) Fig. 2D upper right the red inhibition should point toward the arrow from I to C. An argument is missing why the qualitative behaviour (existence of oscillation), that was obtained from linearized F_b , should persist if the positive part of F_b is used, as was done. Since the mean value \bar{x}_o is close to the zero of F_b , this becomes rather important. The CLE analysis can handle negative rates but changing to a non-linear F_b (by taking the positive part) may alter the behaviour.

Answer: As can be seen from the plots of single-cell trajectories in Figure 2, the copy-number for the output is around 100 for IFF and around 70 for NFB. In our opinion, these copy-numbers are not very high and the dynamics shows significant randomness that is evident from the trajectory plots.

We have corrected the red arrow in Figure 2D. Thanks for pointing this out.

As mentioned in our answer to the previous point, the linearisation is not around $\bar{x}_o = 0$ but around some arbitrary point. For our choice of parameters the linearisation provides an accurate result because the linearised function is very rarely negative for the states visited by the dynamics. We mention this on page 17 in the revised manuscript.

7. *Minor objections in the first part:* - The statement that an SDE approximation is only good for finite time (p.3, lines 8-10), not at stationarity, needs further explanation.
 - (3) is the adjoint operator
 - Eq (35): Why is there a reference to the Cauchy distribution? $S_X(\omega)$ is not a density of a measure.

Answer: The validity of SDE approximations (like the Linear Noise Approximation (LNA) or the Chemical Langevin Equation (CLE)) has only been mathematically established over compact time-intervals like $[0, T]$ (see [12]). Hence even if the conditions for the approximation hold (i.e. all species are in large numbers) the steady-state behavior of the SDE approximation may be different from that of the CTMC model. This is evident from the newly added example of noise-induced oscillation (see response to Reviewer 1) in the antithetic network. Here the deterministic model would exhibit convergence to a fixed point while the CTMC model exhibits sustained oscillations as seen from the off-zero peak in the PSD. In this scenario, the LNA would fail to accurately estimate the PSD of the CTMC model as it basically adds a Gaussian term to the dynamics around the macroscopic process evolving according to the deterministic model. We make this comment when we discuss this noise-induced example (see page 20).

The operator in equation (3) may be called the adjoint operator in some contexts, but in the Markov process theory it is called the generator and therefore we use this terminology in this paper.

$S_X(\omega)$ is indeed the density of a measure of the power at various frequency components. If we divide it by the total area (which is equal to π times the output variance) then we get a probability density, which is Cauchy for the birth-death model. This is interesting

because this distribution has infinite mean and variance, which shows that even for this very simple reaction network, the frequency components in the single-cell trajectory are distributed over a very wide range. We make this comment in the revised manuscript.

8. *As far as the Pade PSD part is concerned, I am neither familiar with the Padé approximation nor with the current state of research on PSD approximations. From that perspective, the approach is new to me and seems innovative. 5.3 and 5.4. discuss interesting applications in synthetic biology, where the method can help to identify parameter regimes that are predicted to yield desired outcomes (stabilized oscillations or oscillation reduction) when used in experiments.*

Answer: We thank the reviewer for these positive comments. In this study we demonstrate how the numerical technique of Padé approximation can be successfully used for PSD estimation. The applications in synthetic biology that we mention in the paper were our main motivation in developing the PSD estimation method.

9. *The authors do not clearly discuss drawbacks of the method. Vaguely, high computational costs are mentioned in the discussion leaving room for improvement.*

Answer: Our method relies on simulating trajectories of a reaction network formed by suitably augmenting the original reaction network. The well-known drawbacks associated with simulating stochastic trajectories are inherited by our method. We now mention this explicitly in the Conclusion section. There are ways to address these drawbacks and make simulations less computationally demanding and we mention some of these approaches in the Conclusion. Essentially any method that enhances simulation efficiency and the estimation accuracy of expectations would help in improving our method. There are a whole host of these methods but trying them with our PSD estimation method is beyond the scope of the current paper.

References

- [1] Chen Jia and Ramon Grima. Frequency domain analysis of fluctuations of mrna and protein copy numbers within a cell lineage: theory and experimental validation. *Physical Review X*, 11(2):021032, 2021. 4, 3
- [2] Laurent Potvin-Trottier, Scott Luro, and Johan Paulsson. Microfluidics and single-cell microscopy to study stochastic processes in bacteria. *Current opinion in microbiology*, 43:186–192, 2018. 1
- [3] Thomas M Norman, Nathan D Lord, Johan Paulsson, and Richard Losick. Memory and modularity in cell-fate decision making. *Nature*, 503(7477):481–486, 2013. 1
- [4] B. Munsky and M. Khammash. The finite state projection algorithm for the solution of the chemical master equation. *Journal of Chemical Physics*, 124(4), 2006. 1
- [5] Brian Munsky, Brooke Trinh, and Mustafa Khammash. Listening to the noise: random fluctuations reveal gene network parameters. *Molecular systems biology*, 5(1):318, 2009. 1

- [6] Brian Munsky, Gregor Neuert, and Alexander Van Oudenaarden. Using gene expression noise to understand gene regulation. *Science*, 336(6078):183–187, 2012. 1
- [7] Gregor Neuert, Brian Munsky, Rui Zhen Tan, Leonid Teytelman, Mustafa Khammash, and Alexander Van Oudenaarden. Systematic identification of signal-activated stochastic gene regulation. *Science*, 339(6119):584–587, 2013. 1
- [8] Sant Kumar, Marc Rullan, and Mustafa Khammash. Rapid prototyping and design of cybergenetic single-cell controllers. *Nature communications*, 12(1):1–13, 2021. 1, 1
- [9] Patrick B Warren, Sorin Tănase-Nicola, and Pieter Rein ten Wolde. Exact results for noise power spectra in linear biochemical reaction networks. *The Journal of chemical physics*, 125(14):144904, 2006. 2
- [10] Philipp Thomas, Hannes Matuschek, and Ramon Grima. How reliable is the linear noise approximation of gene regulatory networks? *BMC genomics*, 14(4):1–15, 2013. 2
- [11] Sanggeun Song, Gil-Suk Yang, Seong Jun Park, Sungguan Hong, Ji-Hyun Kim, and Jaeyoung Sung. Frequency spectrum of chemical fluctuation: A probe of reaction mechanism and dynamics. *PLoS computational biology*, 15(9):e1007356, 2019. 3
- [12] Thomas G Kurtz. Strong approximation theorems for density dependent markov chains. *Stochastic Processes and their Applications*, 6(3):223–240, 1978. 7

Reviewers' Comments:

Reviewer #1:

Remarks to the Author:

The authors have made substantial revisions to the manuscript which address all my questions and comments. The paper has been greatly improved both in its scope of application and in clarifying the detailed computations behind the method. The authors have produced a definitive work on the extraction and use of information contained within the frequency spectra of cellular noise. This will be an important paper in the field of stochastic gene expression and hence I strongly support its publication in Nature Communications.

Reviewer #2:

Remarks to the Author:

The authors addressed all my questions and concerns.

Response Letter

Frequency Spectra and the Color of Cellular Noise

Ankit Gupta^{*1} and Mustafa Khammash^{†1}

¹Department of Biosystems Science and Engineering, ETH Zürich, Mattenstrasse 26, 4058 Basel, Switzerland.

May 31, 2022

We thank the reviewers of *Nature Communications* for going through our paper again. Below are our responses to their comments.

Response to Reviewer 1

1. *The authors have made substantial revisions to the manuscript which address all my questions and comments. The paper has been greatly improved both in its scope of application and in clarifying the detailed computations behind the method. The authors have produced a definitive work on the extraction and use of information contained within the frequency spectra of cellular noise. This will be an important paper in the field of stochastic gene expression and hence I strongly support its publication in Nature Communications.*

Answer: We thank the reviewer for strongly supporting our paper for publication.

Response to Reviewer 2

1. *The authors addressed all my questions and concerns.*

Answer: We thank the reviewer for confirming that his/her concerns have been addressed.

*ankit.gupta@bsse.ethz.ch

†mustafa.khammash@bsse.ethz.ch